# Nuclear translocation of SIRT4 mediates deacetylation of U2AF2 to modulate renal fibrosis through alternative splicing-mediated upregulation of CCN2

**Guangyan Yang**[1,2†]**, Jiaqing Xiang**[1,2†]**, Xiaoxiao Yang**[3]**, Xiaomai Liu**[1,2]**, Yanchun Li**[1,2]**, Lixing Li**[1,2]**, Lin Kang**[1,2]***, Zhen Liang**[1,2]***, Shu Yang**[1,2]*****

[1]Department of Geriatrics, Shenzhen People's Hospital (The Second Clinical Medical College, Jinan University; The First Affiliated Hospital, Southern University of Science and Technology), Guangdong, China; [2]Guangdong Provincial Clinical Research Center for Geriatrics, Shenzhen Clinical Research Center for Geriatrics, Shenzhen People's Hospital (The Second Clinical Medical College, Jinan University; The First Affiliated Hospital, Southern University of Science and Technology), Guangdong, China; [3]Anhui Provincial International Science and Technology Cooperation Base for Major Metabolic Diseases and Nutritional Interventions, Key Laboratory of Metabolism and Regulation for Major Diseases of Anhui Higher Education Institutes, College of Food and Biological Engineering, Hefei University of Technology, Hefei, China

**\*For correspondence:**
Kang.lin@szhospital.com (LK);
liang.zhen@szhospital.com (ZL);
yang.shu@szhospital.com (SY)

[†]These authors contributed equally to this work

**Competing interest:** The authors declare that no competing interests exist.

## eLife Assessment

This study demonstrates a novel role for SIRT4; a mitochondrial deacetylase, shown to translocate into nuclei where it regulates RNA alternative splicing by modulating U2AF2 and the gene expression of CCN2 in tubular cells in response to TGF-β. This **fundamental** work substantially advances our understanding of kidney fibrosis development and offers a potential therapeutic approach. The evidence supporting the conclusions of a SIRT4-U2AF2-CCN2 axis activated by TGF-β is **compelling** and adds a new layer of complexity to the pathogenesis of chronic kidney disease.

**Abstract** TGF-β stimulates CCN2 expression which in turn amplifies TGF-β signaling. This process promotes extracellular matrix production and accelerates the pathological progression of fibrotic diseases. Alternative splicing plays an important role in multiple disease development, while U2 small nuclear RNA auxiliary factor 2 (U2AF2) is an essential factor in the early steps of pre-mRNA splicing. However, the molecular mechanism underlying abnormal *CCN2* expression upon TGF-β stimulation remains unclear. This study elucidates that SIRT4 acts as a master regulator for CCN2 expression in response to TGF-β by modulating U2AF2-mediated alternative splicing. Analyses of renal biopsy specimens from patients with CKD and mouse fibrotic kidney tissues revealed marked nuclear accumulation of SIRT4. The tubulointerstitial fibrosis was alleviated by global deletion or tubular epithelial cell (TEC)-specific knockout of *Sirt4*, and aggravated by adeno-associated virus-mediated SIRT4 overexpression in TECs. Furthermore, SIRT4 was found to translocate from the mitochondria to the cytoplasm through the BAX/BAK pore under TGF-β stimulation. In the cytoplasm, TGF-β activated the ERK pathway and induced the phosphorylation of SIRT4 at Ser36, which further promoted its interaction with importin α1 and subsequent nuclear translocation. In the nucleus, SIRT4 was found to deacetylate U2AF2 at K413, facilitating the splicing of CCN2 pre-mRNA to promote CCN2 protein expression. Importantly, exosomes containing anti-SIRT4 antibodies were

found to effectively mitigate the UUO-induced kidney fibrosis in mice. Collectively, these findings indicated that SIRT4 plays a role in kidney fibrosis by regulating CCN2 expression via the pre-mRNA splicing.

## Introduction

In the kidney and other organ systems, the overexpression (OE) of cellular communication network 2 (CCN2), also known as connective tissue growth factor, is widely recognized as a marker of fibrotic activity (*Gupta et al., 2000*). Transforming growth factor-β (TGF-β) is a master regulator of tissue growth, regeneration, remodeling, and fibrosis, Most TGF-β responses involve CCN2 stimulation at some level, such as the stimulation of extracellular matrix (ECM) components and fibrosis (*Kinashi et al., 2017*; *Tampe and Zeisberg, 2014*). Numerous signaling molecules are involved in the crosstalk and integration of TGF-β and CCN2 effects and vary depending on the cell type and the physiological or pathological process involved. For instance, TGF-β-stimulated SMADs are necessary for the induction of CCN2 expression in normal fibroblasts and basal CCN2 induction in scleroderma fibroblasts, while the maintenance of CCN2 expression is independent of SMADs (*Holmes et al., 2001*). However, it remains unknown whether TGF-β1 regulates CCN2 expression via non-transcriptional pathways.

Sirtuins (Sirts), mammalian Sir2 orthologs, are a highly conserved family of nicotinamide adenine dinucleotide-dependent protein deacetylases and act as important regulators of the aging process, inflammation, cancer, and metabolic diseases (*Morigi et al., 2018*; *Chalkiadaki and Guarente, 2012*; *Baur et al., 2012*). Among the seven known mammalian Sirts, SIRT4 possesses ADP-ribosyltransferase, lipoamidase, and deacylase activities (*Haigis et al., 2006*; *Mathias et al., 2014*). The inhibition of SIRT4 expression has been shown to increase the fat-oxidation capacity of the liver and mitochondrial function in the muscle (*Nasrin et al., 2010*). As loss of fatty acid β-oxidation in the proximal tubule is a critical mediator of acute kidney injury and eventual fibrosis (*Kang et al., 2015*; *Gu et al., 2022*), we hypothesize that SIRT4 may act as a pro-fibrotic factor. However, other studies have demonstrated that SIRT4 OE inhibits glutamine metabolism (*Csibi et al., 2013*), which is necessary for collagen protein synthesis (*Stegen et al., 2019*). This suggests a potential protective role of SIRT4 in the context of fibrosis. However, the specific mechanism by which SIRT4 regulates renal fibrosis remains unclear. Therefore, research is needed to further explore the role and mechanisms of SIRT4 in renal fibrosis, as well as potential therapeutic strategies.

An estimated 60% of all human genes undergo alternative splicing, a highly regulated process that produces splice variants with different functions (*Sazani and Kole, 2003*). There are five small nuclear ribonucleoproteins (snRNPs)—U1, U2, U5, and U4/U6. The status of these proteins helps in maintaining sufficient proteomic diversity for the functional requirements of cell fates and body homeostasis (*Maniatis and Tasic, 2002*). U2 small nuclear RNA auxiliary factor 1 (U2AF1), together with U2AF2, forms the U2AF complex that recognizes and binds to the 3′ splice site of pre-mRNA and recruits U2 snRNPs, thereby facilitating the assembly of the spliceosome, a large RNA-protein complex responsible for the splicing of pre-mRNA (*Zorio and Blumenthal, 1999*; *Wu et al., 1999*; *Merendino et al., 1999*; *Agrawal et al., 2016*). The SF3B complex, which is the core of the U2 snRNP, comprises SF3B1, SF3B2, SF3B3, SF3B4, SF3B5, SF3B6, and PHF5A/ SF3B14b (*Will et al., 2002*). Dysregulation of U2AF2 can lead to the disruption of splicing events, which may contribute to the development and progression of kidney fibrosis. Accordingly, understanding the role of U2AF2 in kidney fibrosis may provide valuable insights into its underlying molecular mechanisms.

Here, we demonstrated that TGF-β1 induced SIRT4 nuclear translocation, resulting in the deacetylation of U2AF2 and recruitment of U2 snRNP, subsequently contributing to pre-mRNA splicing and enhanced protein expression of CCN2. Overall, our study revealed that SIRT4 inhibition can alleviate the progression of renal fibrosis by suppressing CCN2 expression.

## Results

### Nuclear localization of SIRT4 increases in fibrotic kidney

The application of mitochondrial SIRT4 tripartite abundance reporter, a tripartite probe for visualizing the distribution of SIRT4 between mitochondria and the nucleus in single cells (*Ramadani-Muja et al., 2019*), proved the importation of SIRT4 into the mitochondrial matrix and demonstrated its

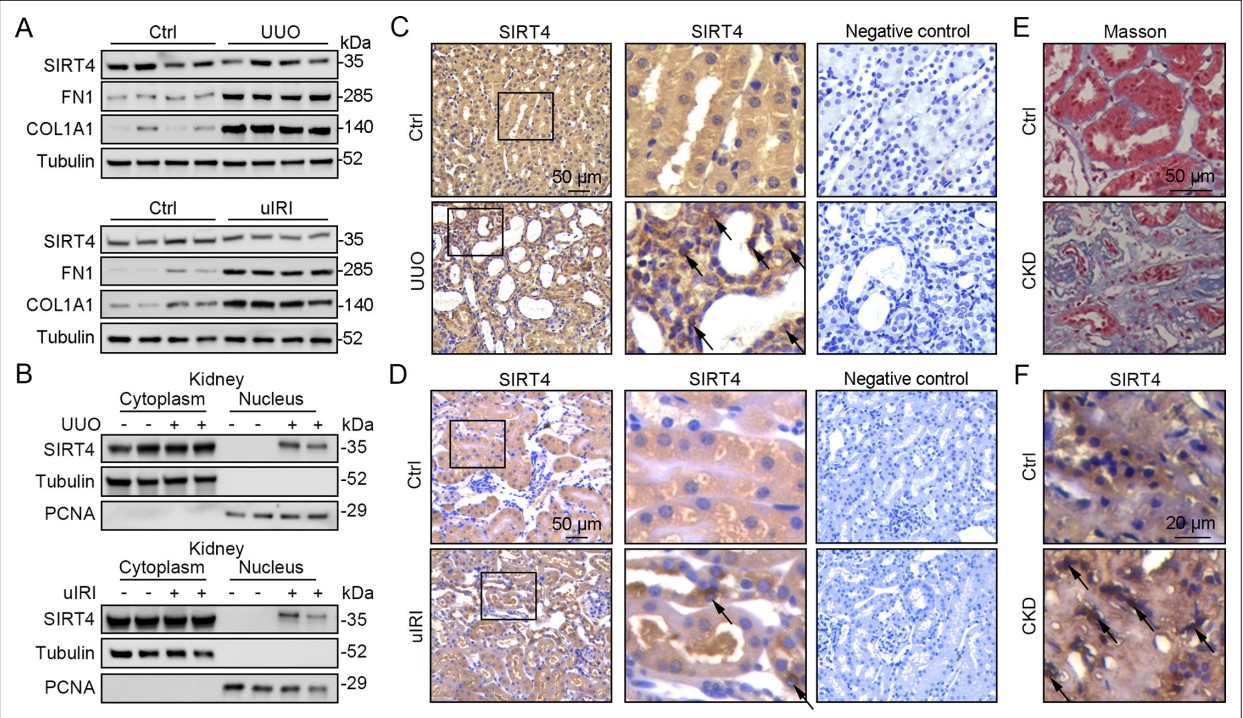

**Figure 1.** Nuclear accumulation of SIRT4 is increased in renal tubules after injury. (**A**) Western blots analysis of SIRT4, FN1, COL1A1, and Tubulin in the kidney of UUO, uIRI, and sham mice (control group). (**B**) Nuclear fractions were prepared from the kidney of UUO, uIRI, and sham mice. Nuclear PCNA and cytoplasmic tubulin were used as controls. (**C, D**) Representative images of immunohistochemical staining of SIRT4 (scale bar, 50 μm) in the kidneys from mice that underwent sham surgery, UUO surgery on day 10 post surgery or uIRI surgery on day 28 post surgery. (**E, F**) Representative images of Masson's trichrome staining (the upper panel; scale bar = 50 μm) and SIRT4 immmohistochemical staining (the bottom panel; scale bar = 20 μm) in the kidney sections from patients with CKD (n=8) and minimal change disease (control group, n=1).

The online version of this article includes the following source data for figure 1:

**Source data 1.** Original files for western blot analysis displayed in *Figure 1A, B*.

**Source data 2.** The uncropped gels or blots with the relevant bands clearly labeled in *Figure 1A and B*.

localization in the nucleus under mitochondrial stress conditions. Nuclear accumulation of SIRT4 was observed in the kidneys following unilateral ureteral occlusion (UUO) or unilateral renal ischemia-reperfusion injury (uIRI) surgery (*Figure 1A–D*). Consistently, elevated nuclear accumulation of SIRT4 was observed in kidney sections from patients with chronic kidney disease (CKD) with severe collagen deposition (*Figure 1E and F*).

## Global deletion of *Sirt4* protects against kidney fibrosis

To determine the role of SIRT4 in kidney fibrosis development in vivo, wild-type (WT) and *Sirt4* global knockout (S4KO) mice were subjected to UUO, uIRI, and folic acid (FA) treatment. Sirius red and Masson's trichrome staining of the kidney sections revealed extensive renal fibrosis in WT mice following UUO (*Figure 2A*), uIRI (*Figure 2E*), and FA administration (*Figure 2—figure supplement 1B*). In contrast, the global deletion of *Sirt4* resulted in a remarkable reduction in the extent of renal fibrosis in all three kidney injury models (*Figure 2A and E* and *Figure 2—figure supplement 1B*). Immunoblots of whole-kidney tissue lysates showed that UUO, uIRI, and FA treatment led to induced levels of the markers of kidney fibrosis (CCN2, FN1, COL1A1, COL3A1, and α-SMA), while the expression of E-cadherin, a hallmark of epithelial-mesenchymal transition, was decreased, with the changes being more pronounced in WT mice than in S4KO mice (*Figure 2B and F* and *Figure 2—figure supplement 1A*). Consistent with the attenuated post-injury fibrotic response in S4KO mouse kidneys observed via imaging studies, deletion of *Sirt4* also mitigated the upregulation of the profibrotic genes *Col1a1*, *Fn1*, *Eln*, *Ccn2*, *Acta2*, and *Col3a1* (*Figure 2C and G* and *Figure 2—figure supplement 1C*). In addition, the mRNA levels of both neutrophil gelatinase-associated lipocalin (*Ngal*) and

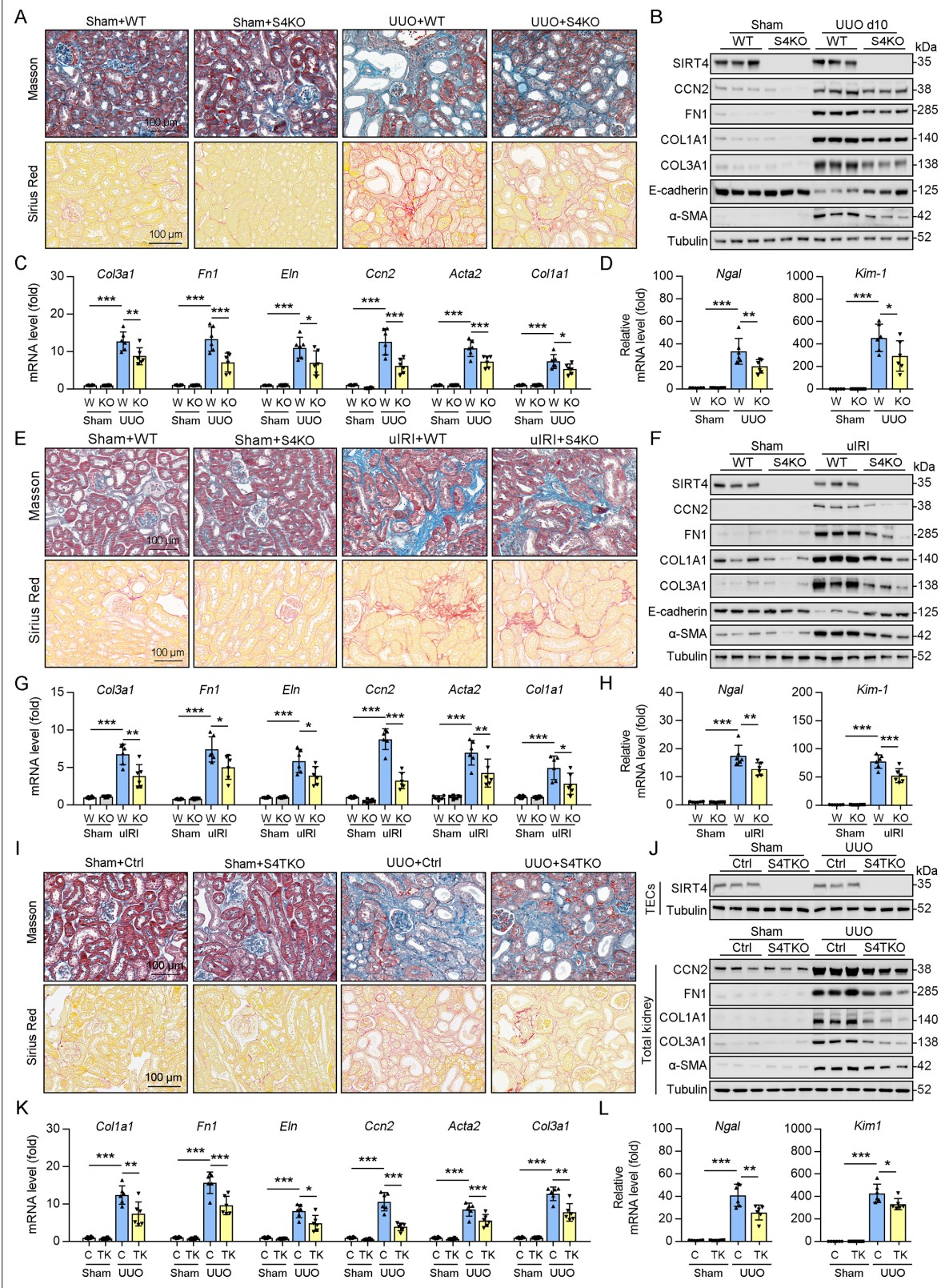

**Figure 2.** Knockout of *Sirt4* or targeted deletion of *Sirt4* in renal TECs alleviates renal fibrosis induced by UUO or uIRI. (**A–D**) WT or S4KO mice were randomly assigned to sham or UUO surgery according to an established protocol. Kidney samples were obtained from mice on day 10 post UUO or sham surgery (n=6 per group). (**A**) Representative images of Masson's trichrome staining and Sirius red in kidneys from mice (scale bar = 100 µm). (**B**) Western blot analysis of the expression of SIRT4, CCN2, FN1, COL1A1, COL3A1, E-cadherin, α-SMA, and Tubulin in the kidney of mice. (**C**, **D**) The

*Figure 2 continued on next page*

*Figure 2 continued*

mRNA level of *Col1a1, Fn1, Eln, Ccn2, Acta2, Col3a1, Ngal*, and *Kim-1* in the kidney of mice. (**E–H**) WT or S4KO mice received uIRI surgery, and the sham surgery kidneys were used as control. Kidney samples collected from mice after surgery at 28 d (n=6 per group). (**E**) Representative images of Masson's trichrome staining and Sirius red in kidneys from mice (scale bar = 100 μm). (**F**) Western blot analysis of the expression of SIRT4, CCN2, FN1, COL1A1, COL3A1, E-cadherin, α-SMA, and Tubulin in the kidney of mice. (**G, H**) The mRNA level of *Col1a1, Fn1, Eln, Ccn2, Acta2, Col3a1, Ngal*, and *Kim-1* in the kidney of mice. (**I–L**) WT or S4TKO mice were randomly assigned to sham or UUO surgery according to an established protocol. Kidney samples were obtained from mice on day 10 post UUO or sham surgery (n=6 per group). (**I**) Representative images of Masson's trichrome staining and Sirius red in kidneys from mice (scale bar = 100 μm). (**J**) Western blot analysis of the expression of SIRT4, CCN2, FN1, COL1A1, COL3A1, α-SMA, and Tubulin in the kidney of mice. (**K, L**) The mRNA level of *Col1a1, Fn1, Eln, Ccn2, Acta2, Col3a1, Ngal*, and *Kim-1* in the kidney of mice. For all panels, data are presented as mean ± SD. *p<0.05, **p<0.01, ***p<0.001 by one-way ANOVA with Bonferroni correction test.

The online version of this article includes the following source data and figure supplement(s) for figure 2:

**Source data 1.** Original files for western blot analysis displayed in *Figure 2B, F and J*.

**Source data 2.** The uncropped gels or blots with the relevant bands clearly labeled in *Figure 2B, F and J*.

**Figure supplement 1.** Knockout of SIRT4 aggravated FA-induced kidney fibrosis.

**Figure supplement 1—source data 1.** Original files for western blot analysis displayed in *Figure 2—figure supplement 1A, E, F*.

**Figure supplement 1—source data 2.** The uncropped gels or blots with 1109 the relevant bands clearly labeled in *Figure 2—figure supplement 1A, E, F*.

---

kidney injury molecule 1 (*Kim-1*), which are markers of acute kidney injury, were significantly increased in the kidney tissues from WT and S4KO mice following injury (**Figure 2D and H** and **Figure 2—figure supplement 1D**). Notably, the levels of *Ngal* and *Kim-1* were significantly reduced in S4KO mice compared to those in WT mice following injury (**Figure 2D and H** and **Figure 2—figure supplement 1D**). Collectively, these results suggest that *Sirt4* deletion protects mice against renal fibrosis.

## Deletion of *Sirt4* in renal tubule epithelial cells markedly attenuates the extent of kidney fibrosis following injury

Using conventional agarose gel-based RT-PCR and western blot analysis, we analyzed the expression of SIRT4 in renal parenchymal cells, including mouse podocytes (MPCs), glomerular endothelial cells (GECs), kidney fibroblasts (KFs), and tubule epithelial cells (TECs). Compared to their levels in TECs, the basal levels of SIRT4 in MPCs and GECs were relatively low, while in KFs is moderate (**Figure 2—figure supplement 1E**). To determine the in vivo contribution of SIRT4 in renal TECs and fibroblasts to the development of kidney fibrosis, we generated TEC-specific (S4TKO; Cdh16-cre/ERT2×*Sirt4*^flox/flox^) and fibroblast-specific (S4FKO; Col1a2-Cre/ERT2×*Sirt4*^flox/flox^) *Sirt4* knockout mice; in these experiments, Cadh16-cre mice and tamoxifen-treated Col1a2-Cre mice were used as controls, respectively. The extent of kidney fibrosis induced by UUO was markedly reduced in S4TKO mice compared to that in control mice (**Figure 2I and J**). In addition, the expression levels of kidney fibrosis markers (CCN2, FN1, COL1A1, COL3A1, and α-SMA) were also significantly reduced in S4TKO compared to those in control mice following UUO surgery (**Figure 2J**). Consistently, the profibrotic genes (*Col1a1, Fn1, Eln, Ccn2, Acta2*, and *Col3a1*) and acute kidney injury markers *Nagl* and *Kim-1* were downregulated in S4TKO mice compared to those in control mice (**Figure 2K and L**). Nevertheless, the targeted deletion of SIRT4 in fibroblasts did not affect the extent of UUO-induced kidney fibrosis (**Figure 2—figure supplement 1F–H**). These results suggest that SIRT4 expressed in TECs but not in fibroblasts primarily contributes to the pathogenesis of kidney fibrosis.

## SIRT4 OE aggravates kidney fibrosis

Since the deletion of *Sirt4* attenuated the post-injury fibrotic response, we next determined whether *Sirt4* OE can aggravate the response. We introduced SIRT4 into WT mouse kidneys using the adeno-associated virus serotype 9 vector with a tubule-specific *Ksp-cadherin* promoter, and AAV9-*Ksp-null* transfection was used as the control treatment. Targeting *Sirt4* OE in kidney TECs markedly aggravated the extent of kidney fibrosis induced by UUO, uIRI, and FA treatment compared to that in the control mice (**Figure 3A, D and G**). The expression levels of kidney fibrosis markers (CCN2, FN1, COL1A1, COL3A1, and α-SMA) and the decreased E-cadherin were also significantly enhanced in *Sirt4* OE mice compared to those in the control mice following kidney injury (**Figure 3B, E and H**). Consistently, *Sirt4* OE upregulated the transcription of profibrotic genes, namely *Col1a1, Fn1, Eln*,

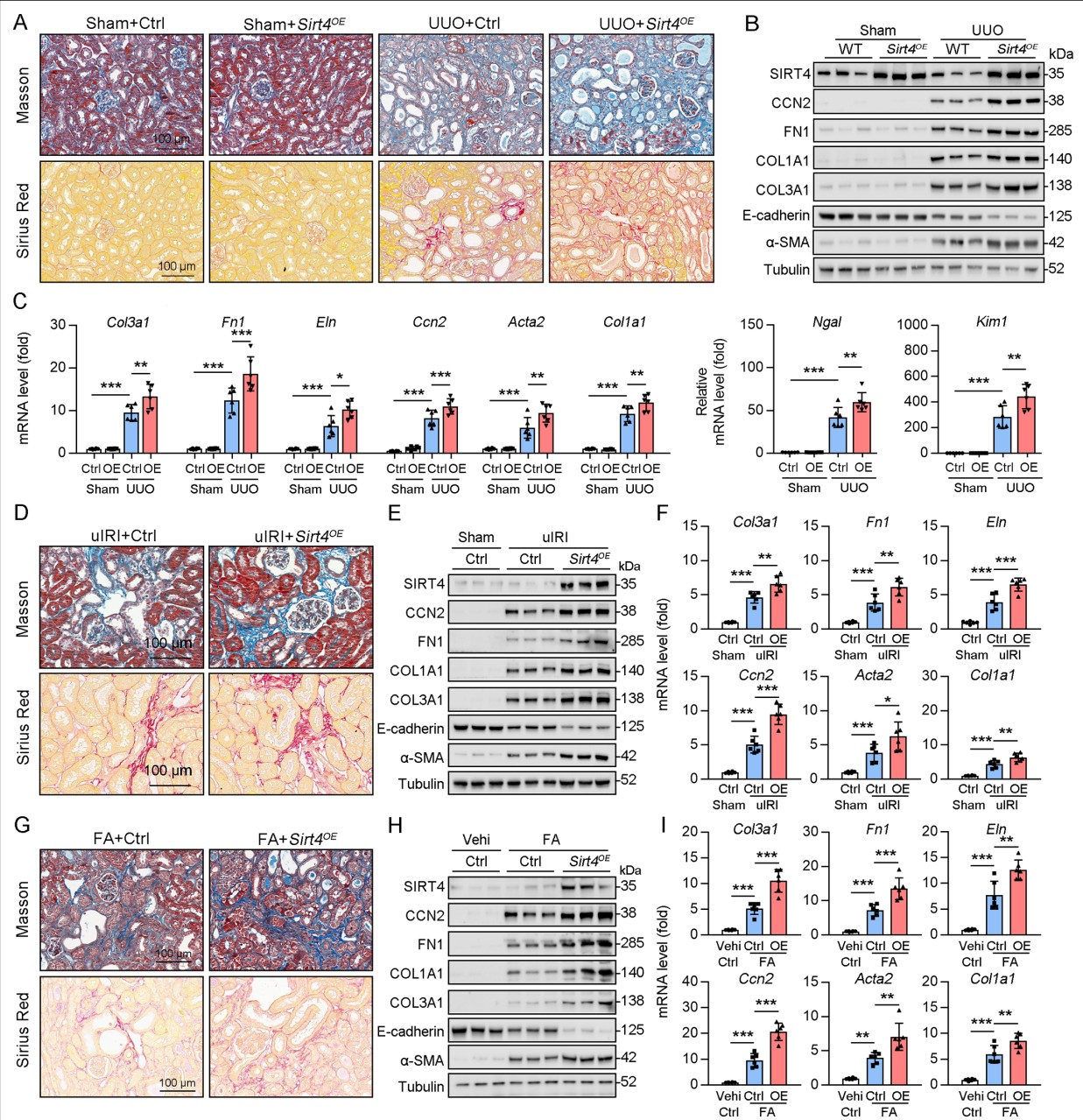

**Figure 3.** TECs specific SIRT4[OE] aggravates kidney fibrosis. (**A–I**) AAV9-Ctrl or AAV9-*Ksp-Sirt4* was injected into the kidneys of mice in situ at three independent points. After 2-week transfection, the mice received UUO surgery, uIRI surgery, or FA treatment (vehicle treatment as control) (n=6 per group). (**A, D, G**) Representative images of Masson's trichrome staining and Sirius red in kidneys from mice (scale bar = 100 μm). (**B, E, H**) Western blot analysis of the expression of SIRT4, CCN2, FN1, COL1A1, COL3A1, E-cadherin, α-SMA, and Tubulin in the kidney of mice. (**C, F, I**) The mRNA level of *Col3a1, Fn1, Eln, Ccn2, Acta2, Col1a1, Ngal*, and *Kim-1* in the kidney of mice. For all panels, data are presented as mean ± SD. *p<0.05, **p<0.01, ***p<0.001 by one-way ANOVA with Bonferroni correction test.

The online version of this article includes the following source data and figure supplement(s) for figure 3:

**Source data 1.** Original files for western blot analysis displayed in *Figure 3B, E and H*.

**Source data 2.** The uncropped gels or blots with the relevant bands clearly labeled in *Figure 3B, E and H*.

**Figure supplement 1.** TECs targeted overexpression of SIRT4 reverses S4KO reduced kidney fibrosis in UUO mice.

**Figure supplement 1—source data 1.** Original files for western blot analysis displayed in *Figure 3—figure supplement 1A*.

**Figure supplement 1—source data 2.** The uncropped gels or blots with the relevant bands clearly labeled in *Figure 3—figure supplement 1A*.

*Ccn2, Acta2*, and *Col3a1*, in mouse kidneys following injury, compared to that in the control group (*Figure 3C, F, I*). Further analysis revealed an evident upregulation of *Ngal* and *Kim-1* transcripts in the kidneys of *Sirt4* OE mice compared to that in the kidneys of control mice in response to UUO (*Figure 3C*). Notably, targeting WT *Sirt4* OE in kidney TECs markedly reversed the decline in collagen deposition and the reduction in ECM-related protein expression in S4KO mice (*Figure 3—figure supplement 1A–C*). These results demonstrate that interventional SIRT4 OE in renal TECs exacerbates kidney fibrosis progression.

## SIRT4 interacts with U2AF2 under TGF-β stimulation

To explore the action mechanism of SIRT4 in kidney fibrosis, we first identified the proteins interacting with SIRT4 in TECs. We employed rapid immunoprecipitation and mass spectrometry of endogenous proteins (RIME), which is an efficient and unbiased proteomic approach for identifying interacting proteins. Human TECs were treated with TGF-β1 or the control (DMSO) for 24 hr, followed by protein-DNA crosslinking in 1% formaldehyde. Cells were sonicated, followed by immunoprecipitation with an SIRT4 antibody (*Figure 4A*). Mass spectrometry identified a total of 1081 unique proteins that copurified with SIRT4. We only considered SIRT4-associated proteins that occurred in three of three independent replicates and excluded any proteins that appeared in any one of the IgG control RIMEs. As a result, we selected of 22 SIRT4-associated proteins. Notably, the SIRT4 interaction involved numerous specific nuclear localization proteins, such as PUF60, U2AF2, RPS2, TSR1, ZC3H15, and SART1. Of these, U2AF2 and PUF60 function cooperatively in pre-mRNA splicing (*Hastings et al., 2007*). The levels (normalized) of PUF60 and U2AF2 are shown in the *Figure 4B*. We next determined whether PUF60 or U2AF2 was recruited to SIRT4 in TECs stimulated with TGF-β1. Immunoblotting of immunoprecipitated SIRT4 with anti-PUF60 and -U2AF2 antibodies showed that TGF-β1 treatment resulted in an interaction among SIRT4, PUF60, and U2AF2 (*Figure 4C*, left panel). Whole-kidney tissue lysates consistently showed an interaction among SIRT4, PUF60, and U2AF2 (*Figure 4C*, middle panel). As U2AF2 has been shown to interact with PUF60 (*Hastings et al., 2007*), we tested whether SIRT4 interacts with either of the two proteins via the U2AF2-PUF60 interaction. The interaction between SIRT4 and PUF60 was abolished in U2af2-knockout cells (*Figure 4D*). However, the interaction of SIRT4 and U2AF2 was not affected in *Puf60* knockdown cells (data not shown). Furthermore, exogenous co-IP revealed an interaction between SIRT4 and U2AF2 (*Figure 4C*, right panel). Together, these results suggest that SIRT4 interacts with U2AF2 under TGF-β1 stimulation or kidney injury.

## U2AF2 acetylation is decreased under cellular stress

Accumulating evidence suggests that SIRT4 can show weak ADP-ribosyltransferase (*Haigis et al., 2006*; *Schwer et al., 2002*; *Ahuja et al., 2007*) as well as substrate-specific deacetylase activity (*Laurent et al., 2013*; *Huang et al., 2023*; *Gu et al., 2020*), akin to that observed for SIRT6 and SIRT7 (*Barber et al., 2012*; *Zhong et al., 2010*). Hence, we tested whether SIRT4 regulates the ADP-ribosylation or acetylation of U2AF2. Our results showed that *Sirt4* OE reduced the levels of acetylated U2AF2 (*Figure 4E*). However, the ADP-ribosylation of U2AF2 was extremely low in both SIRT4 OE and control cells (*Figure 4E*). According to the database Compendium of Protein Lysine Modifications 4.0, lysine (K) 70, 276, 292, 413, 453, and 462 on U2AF2 can be acetylated. Hence, we mutated all six K residues to arginine (R), which mimics the deacetylated state of the protein. As shown in *Figure 4F*, the U2AF2-K413R mutant showed reduced acetylation compared to U2AF2 WT. Conservation analysis of U2AF2 indicated that K413 is a highly conserved site spanning from *Schizosaccharomyces pombe* to *Homo sapiens* (*Figure 4G*). Next, we examined whether U2AF2 K413 is the key acetylation site in response to TGF-β1 stimulation. TGF-β1 treatment reduced the levels of acetylated U2AF2 in WT but had little effect in the K413R mutant U2AF2 (*Figure 4H*). Consistently, U2AF2 acetylation decreased in the kidney following injury (*Figure 4I*). Interestingly, U2AF2 acetylation was also remarkably reduced in the cells subjected to serum starvation, endoplasmic reticulum (ER) stress (induced by tunicamycin), viral infection [poly (I:C)], or DNA damage stress (induced by cisplatin) (*Figure 4J*). These results collectively demonstrate that U2AF2 deacetylation occurs universally in response to cellular stress.

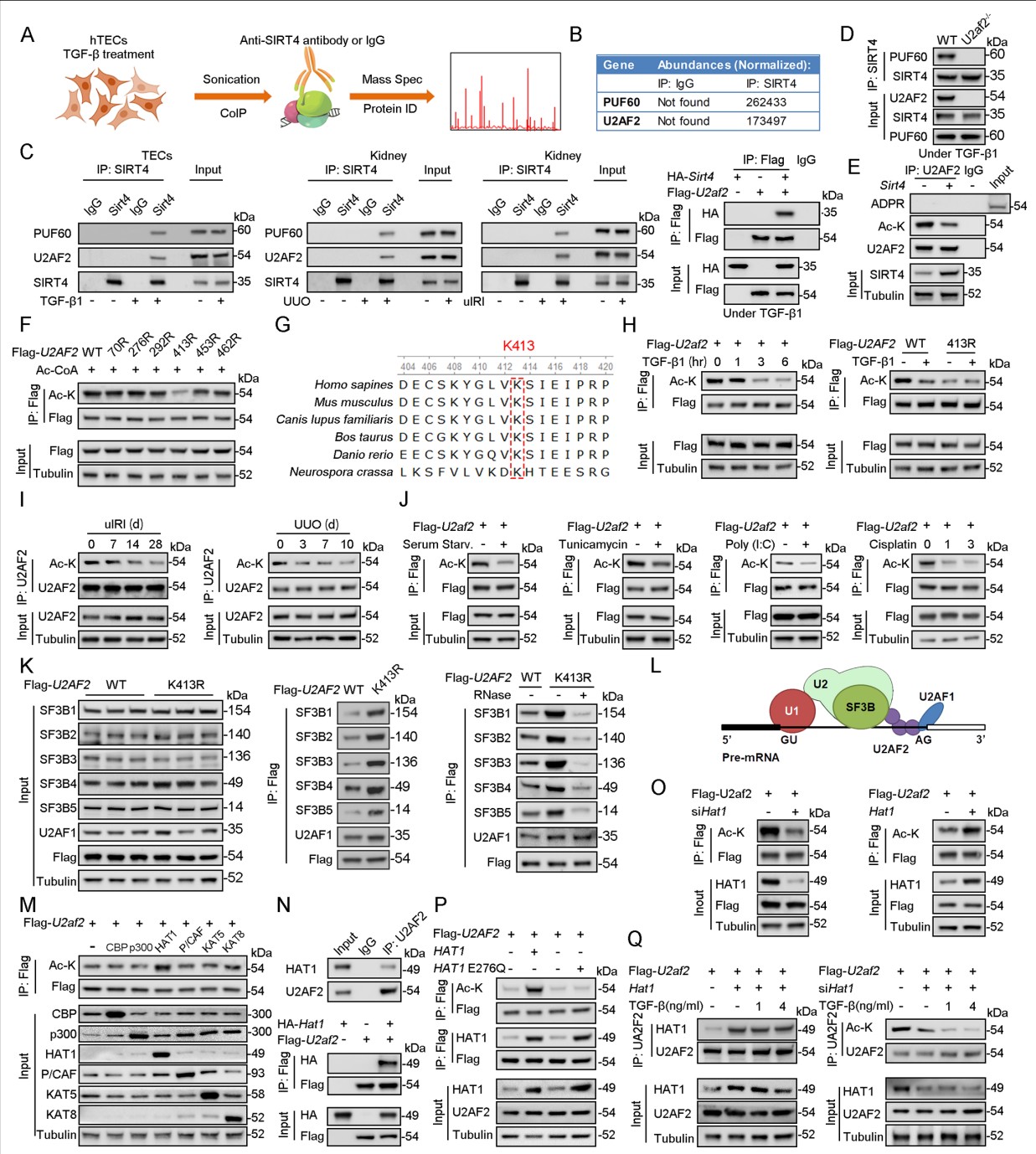

**Figure 4.** U2AF2 acetylation at K413 is increased under TGF-β1 stimulation. (**A**) Proteins interacted with SIRT4 in chromatin were identified by RIME. (**B**) The abundances (Normalized) of PUF60 and U2AF2 were shown. (**C**) Protein lysates from TECs were exposed to TGF-β1(left panel), kidney from the sham, uIRI, and UUO mice (middle panels), and WT TECs transfected with HA-*Sirt4* and/or Flag-*U2af2* and then treated with TGF-β1 (right panel), the protein lysates were subjected to Co-IP with anti-SIRT4 antibody, and determined protein expression using western blotting by the indicated antibodies. (**D**) TECs isolated from WT or *U2af2⁻/⁻* mice and then treated with TGF-β1, cells lysates subjected to Co-IP with anti-SIRT4 antibody, and western blotting using indicated antibodies. (**E**) WT TECs transfected with Ad-*Sirt4* or Ad-null as indicated, and cells lysates subjected to Co-IP with anti-U2AF2 antibody, and western blotting using indicated antibodies. (**F**) HK2 cells (human renal TECs) transfected with Flag- U2AF2 WT, K70R, K276R, K272R, K292R, K413R, K453R, or K462R under Ac-CoA treatment, cells lysates subjected to Co-IP with anti-Flag antibody, and western blotting using indicated antibodies. (**G**) The conservation of U2AF2 lysine 413 in different species. (**H**) TECs transfected with Flag-*U2af2* WT under TGF-β1 stimulation, cells lysates subjected to Co-IP with anti-Flag antibody, and western blotting using indicated antibodies (left panel). HK2 cells transfected with Flag-*U2AF2* WT or K413R with or without TGF-β1, cells lysates subjected to Co-IP with anti-Flag antibody, and western blotting using indicated antibodies (right

*Figure 4 continued on next page*

*Figure 4 continued*

panel). (**I**) Kidney lysates subjected to Co-IP with anti-U2AF2 antibody in uIRI, and UUO mice, and western blotting using indicated antibodies (middle and right panels). (**J**) WT TECs transfected with Flag-*U2af2* WT and then under serum starvation, Tunicamycin, Poly (**I, C**), and Cisplatin stimulation. Cells lysates subjected to Co-IP with anti-Flag antibody, and western blotting using indicated antibodies. (**K**) HK2 cells transfected with Flag-*U2AF2* WT or K413R and cells lysates subjected to Co-IP with anti-Flag antibody, and western blotting using indicated antibodies (left and middle panels). HK2 cells transfected with Flag-*U2AF2* WT or K413R with or without RNase stimulation, cells lysates subjected to Co-IP with anti-Flag antibody, and western blotting using indicated antibodies (right panel). (**L**) Schematic representation of components of SF3B complex performing pre-mRNA BPS recognition. (**M**) WT TECs transfected with Flag-*U2af2* WT with CBP, p300, HAT1, P/CAF, KAT5, or KAT8, cells lysates subjected to Co-IP with anti-Flag antibody, and western blotting using indicated antibodies. (**N**) Whole-kidney lysates were immunoprecipitated with anti-U2AF2 antibody, and precipitated proteins were detected by indicated antibodies (the upper panel). WT TECs was transfected with HA-*Hat1* and/or Flag-*U2af2*. Cells lysates subjected to Co-IP with anti-Flag antibody, and western blotting using indicated antibodies. (**O**) WT TECs was transfected with Flag-*U2af2* WT with Ad-sh*Hat1* or Ad-*Hat1*. Cells lysates subjected to Co-IP with anti-Flag antibody, and western blotting using indicated antibodies. (**P**) HK2 cells were transfected with Flag-*U2AF2* WT, Ad-*HAT1*, or Ad-*HAT1* E276Q as indicated in figure. Cells lysates subjected to Co-IP with anti-Flag antibody, and western blotting using indicated antibodies. (**Q**) WT TECs was transfected with Flag-*U2af2*, Ad-*Hat1* or Ad-sh*Hat1* under TGF-β1 stimulation. Cells lysates subjected to Co-IP with anti-Flag antibody, and western blotting using indicated antibodies.

The online version of this article includes the following source data for figure 4:

**Source data 1.** Original files for western blot analysis displayed in *Figure 4C, D, E, F, H, I, J, K, L, M, N, O, P and Q*.

**Source data 2.** The uncropped gels or blots with the relevant bands clearly labeled in *Figure 4C, D, E, F, H, I, J, K, L, M, N, O, P and Q*.

## U2AF2 acetylation affects the interaction of U2AF and SF3B complexes

Next, we investigate the effect of U2AF2 acetylation at K413 on the formation of the U2AF complex and recruitment of the SF3B complex (*Figure 4K*). The expression of K413R did not regulate the stability of the SF3B and U2AF components. Next, we performed an immunoprecipitation assay to detect the interaction between the U2AF and SF3B complexes using U2AF2-WT- and U2AF2-K413R-expressing cells. Interestingly, U2AF2 K413R interacted strongly with components of the SF3B complex and with U2AF1 compared to that with U2AF2 WT (*Figure 4K*). Notably, the interaction of U2AF2 and U2AF1 was not obviously regulated after RNAase treatment, while the interaction of U2AF2 and SF3B complex was reduced following RNase treatment (*Figure 4K*). These findings implied the interactions between U2AF1 and U2AF2 were independent of RNA binding, while the interaction of U2AF2 and SF3B complex were dependent on RNA binding. As a component of the U2AF complex, U2AF2 interacts with U2AF1 to form the U2AF complex, which recognizes the 3′ splice site (3′ SS) of U2 introns and recruits U2 snRNP (*Zorio and Blumenthal, 1999*; *Figure 4L*). Taken together, these data suggest that U2AF2 acetylation regulates the formation of the U2AF complex and recruitment of the SF3B complex.

## U2AF2 acetylation under cellular stress is not HAT1-dependent

To identify the histone acetyltransferase (HAT) responsible for U2AF2 acetylation under cellular stress, we co-transfected U2AF2 with different HATs, namely CBP, p300, HAT1, P/CAF, KAT5, and KAT8. Of these, HAT1 mainly promoted U2AF2 acetylation (*Figure 4M*). Both endogenous and exogenous co-IP results showed an interaction between HAT1 and U2AF2 (*Figure 4N*). Further experiments were performed to determine whether U2AF2 acetylation in response to stress is dependent on HAT1. We found that knockdown of *Hat1* reduced U2AF2 acetylation, while *Hat1* OE induced the acetylation (*Figure 4O*). Furthermore, U2AF2 acetylation was not affected by enzymatically defective HAT1 (*Figure 4P*). Although U2AF2 interacted with HAT1, the interaction was altered weakly under TGF-β1 stimulation (*Figure 4Q*, left panel). Consistently, under *Hat1* knockdown conditions as well, TGF-β1 stimulation further reduced U2AF2 acetylation (*Figure 4Q*, right panel). Thus, although HAT1 acetylated U2AF2, the decreased acetylation level of U2AF2 by TGF-β1 stimulation was independent of HAT1.

## SIRT4 deacetylates U2AF2 at Lys413

As SIRT4 interacted with U2AF2, we intend to figure out the directly role of SIRT4 in U2AF2 acetylation. We found that SIRT4 OE reduced U2AF2 acetylation, whereas SIRT4 knockdown increased its acetylation (*Figure 5A*). Additionally, TGF-β1 stimulated the interaction between SIRT4 and U2AF2 in a dose-dependent manner (*Figure 5B*). Consistently, the TGF-β1-induced decrease in U2AF2 acetylation was blocked by the knockdown of *Sirt4* (*Figure 5C*). In contrast, SIRT4 OE further enhanced the

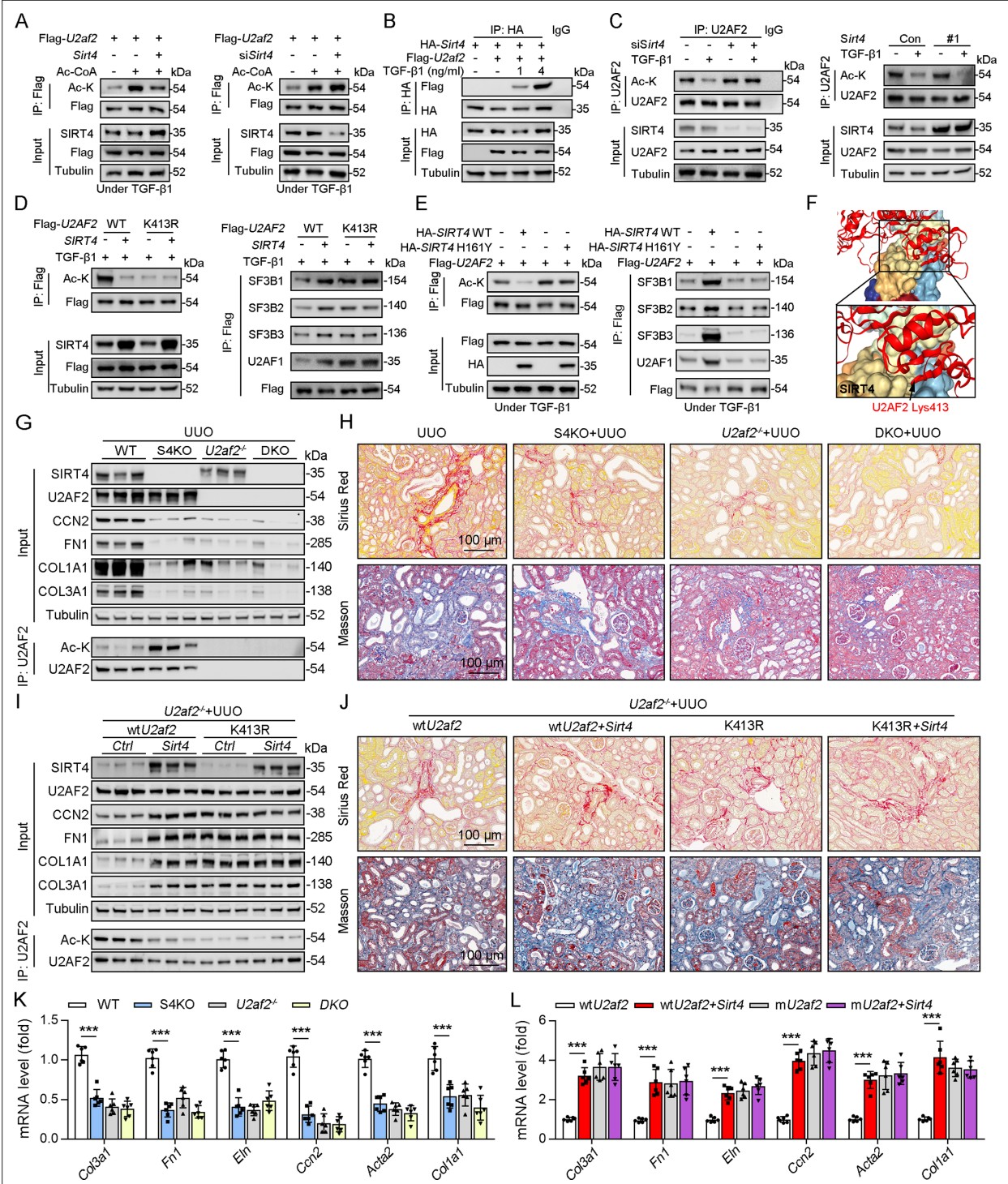

**Figure 5.** U2AF2 deacetylation under TGF-β1 is SIRT4 dependent. (**A**) WT TECs were transfected with Flag-*U2af2* WT, Ad-*Sirt4* or Ad-sh*Sirt4* under Ac-CoA treatment. Cells lysates subjected to Co-IP with anti-Flag antibody, and western blotting using indicated antibodies. (**B**) WT TECs were transfected with Flag-*U2af2* WT and HA-*Sirt4* under TGF-β1 stimulation. Cells lysates subjected to Co-IP with anti-HA antibody and Western blotting using indicated antibodies. (**C**) WT TECs were transfected with Ad-sh*Sirt4* or Ad-*Sirt4* with or without TGF-β1 stimulation. Cells lysates subjected to Co-IP with anti-U2AF2 antibody, and western blotting using indicated antibodies. (**D**) HK2 cells were transfected with Flag-*U2AF2* WT or K413R, Ad-sh*SIRT4* under TGF-β1 stimulation. Cells lysates subjected to Co-IP with anti-Flag antibody, and Western blotting using indicated antibodies. (**E**) HK2 cells were transfected with HA-*SIRT4* WT or H161Y and Flag-*U2AF2* under TGF-β1 treatment. Cells lysates subjected to Co-IP with anti-Flag antibody, and western blotting using indicated antibodies. (**F**) SIRT4-U2AF2 docking with the HDOCK server. High magnification of boxed areas is presented on the left. Arrow

*Figure 5 continued on next page*

*Figure 5 continued*

indicates K413 of U2AF2 protein. (**G, H, K**) WT, S4KO, U2AF2⁻/⁻ or DKO mice were randomly assigned to UUO surgery according to an established protocol. Kidney samples were obtained from mice on day 10 post surgery or sham surgery (n=6 per group). (**G**) Western blot analysis of the expression of SIRT4, U2AF2, CCN2, FN1, COL1A1, COL3A1, and Tubulin in the kidney from mice. Kidney lysates subjected to Co-IP with anti-U2AF2 antibody, and western blotting using indicated antibodies. (**H**) Representative images of Masson's trichrome staining and Sirius red in kidneys from mice (scale bar, 100 μm). (**K**) The mRNA level of *Col1a1, Fn1, Eln, Ccn2, Acta2* and *Col3a1* in the kidney of mice. (**I, J, L**) *U2af2⁻/⁻* mice received in situ renal injection of AAV9-*Ksp-U2af2* WT (*wtU2af2*), *wtU2af2* plus *Sirt4*, AAV9-*Ksp-U2af2* K413R (*mutU2af2*), *K413R* plus *Sirt4* at 6 weeks of age. After 2 weeks, the mice were randomly assigned to sham surgery or UUO surgery according to an established protocol. Kidney samples were obtained from mice on day 10 post surgery or sham surgery (n=6 per group). (**I**) Western blot analysis of the expression of SIRT4, U2AF2, CCN2, FN1, COL1A1, COL3A1, and Tubulin in the kidney from mice. Kidney lysates subjected to Co-IP with anti-U2AF2 antibody, and western blotting using indicated antibodies. (**J**) Representative images of Masson's trichrome staining and Sirius red in kidneys from mice (scale bar, 100 μm). (**L**) The mRNA level of *Col1a1, Fn1, Eln, Ccn2, Acta2, and Col3a1* in the kidney of mice. For all panels, data are presented as mean ± SD. *p<0.05, **p<0.01, ***p<0.001 by one-way ANOVA with Bonferroni correction test.

The online version of this article includes the following source data for figure 5:

**Source data 1.** Original files for western blot analysis displayed in *Figure 5A, B, C, D, E, G, I*.

**Source data 2.** The uncropped gels or blots with the relevant bands clearly labeled in *Figure 5A, B, C, D, E, G I*.

TGF-β1-induced U2AF2 deacetylation (*Figure 5C*). Compared to U2AF2 WT, U2AF2 K413R blocked the SIRT4 OE-induced deacetylation of U2AF2 as well as the interaction of U2AF2 with U2AF1 and the SF3B complex under TGF-β1 stimulation (*Figure 5D*). Accordingly, these phenotypes induced by SIRT4 OE were restricted when an enzyme-defective SIRT4 (SIRT4 H161Y) was used (*Figure 5E*). Molecular docking simulations showed that U2AF2 Lys413 was present at the contact surface between SIRT4 and U2AF2 (*Figure 5F*), further demonstrating that SIRT4 can interact with U2AF2 to promote its deacetylation.

To determine whether S4KO alleviates renal fibrosis by upregulating the acetylation of U2AF2 in mouse kidney, we crossed *Cdh16*-cre/ERT2×*U2af2^flox/flox^* (*U2af2⁻/⁻*) with *Sirt4⁻/⁻* mice to generate DKO mice and then performed UUO surgery. No body weight differences were noted due to genetic manipulation (data not shown). Compared to that in WT mice, the acetylation level of U2AF2 was increased, but renal fibrosis was reduced in S4KO mice after UUO surgery (*Figure 5G, H and K*). The UUO-induced kidney injury was largely reduced in DKO and U2af2⁻/⁻ mice, which showed comparable collagen deposition and expression of ECM-related proteins and mRNA (*Figure 5G, H and K*). We next treated U2af2⁻/⁻ mice with AAV-wt*U2af2* or AAV-m*U2af2* (*U2af2* K413R) to re-express the two types of U2AF2 proteins through in situ renal injection. Mice were followed by administered of AAV-*Sirt4* or AAV-Ctrl and then subjected to UUO surgery. The U2AF2 K413R OE mice showed a greater extent of renal fibrosis than did the wt U2AF2 OE mice, as evidenced by collagen deposition and increased ECM protein levels (*Figure 5I, J and L*). Importantly, the extent of renal fibrosis was remarkably augmented in mice injected with wt*U2af2* OE and *Sirt4* OE compared to that in wt U2AF2 OE mice (*Figure 5I, J and L*). However, *Sirt4* OE had little synergistic effect on mice injected with AAV-m*U2af2* (*Figure 5I, J and L*). Together, these data support that SIRT4-mediated deacetylation of U2AF2 at K413 is an important step in SIRT4-induced renal fibrosis.

## SIRT4 deacetylates U2AF2 at K413 to regulate *CCN2* expression and pre-mRNA splicing

U2AF1 S34F and Q157R mutations have been reported to compromise U2AF2–RNA interactions, resulting predominantly in intron retention and exon exclusion (*Biancon et al., 2022*). Hence, we hypothesized that the SIRT4-mediated deacetylation of U2AF2 participates in the regulation of these processes. RNA-sequencing was performed to identify the changes in gene expression and alternative splicing in adenovirus-mediated SIRT4^OE^-transfected cells or adenovirus-Ctrl-transfected cells after TGF-β1 stimulation (*Figure 6A*). Surprisingly, SIRT4 stimulation upregulated several genes (p<0.01; *Supplementary file 1a*) that are involved in mRNA processing and the TGF-β signaling pathway. A total of 142 differentially expressed genes (p<0.0005) were identified (*Figure 6A*). Meanwhile, 248 genes showed differential intron retention (FDR <0.05; *Supplementary file 1b*; *Figure 6A*). This analysis revealed four genes that were common between the differentially expressed genes and the differentially spliced genes (*Figure 6A*). *Ccn2*, one of these four genes, ranked third among the all upregulated genes (p<0.0005) after SIRT4 overexpression. Given that intron retention due to abnormal

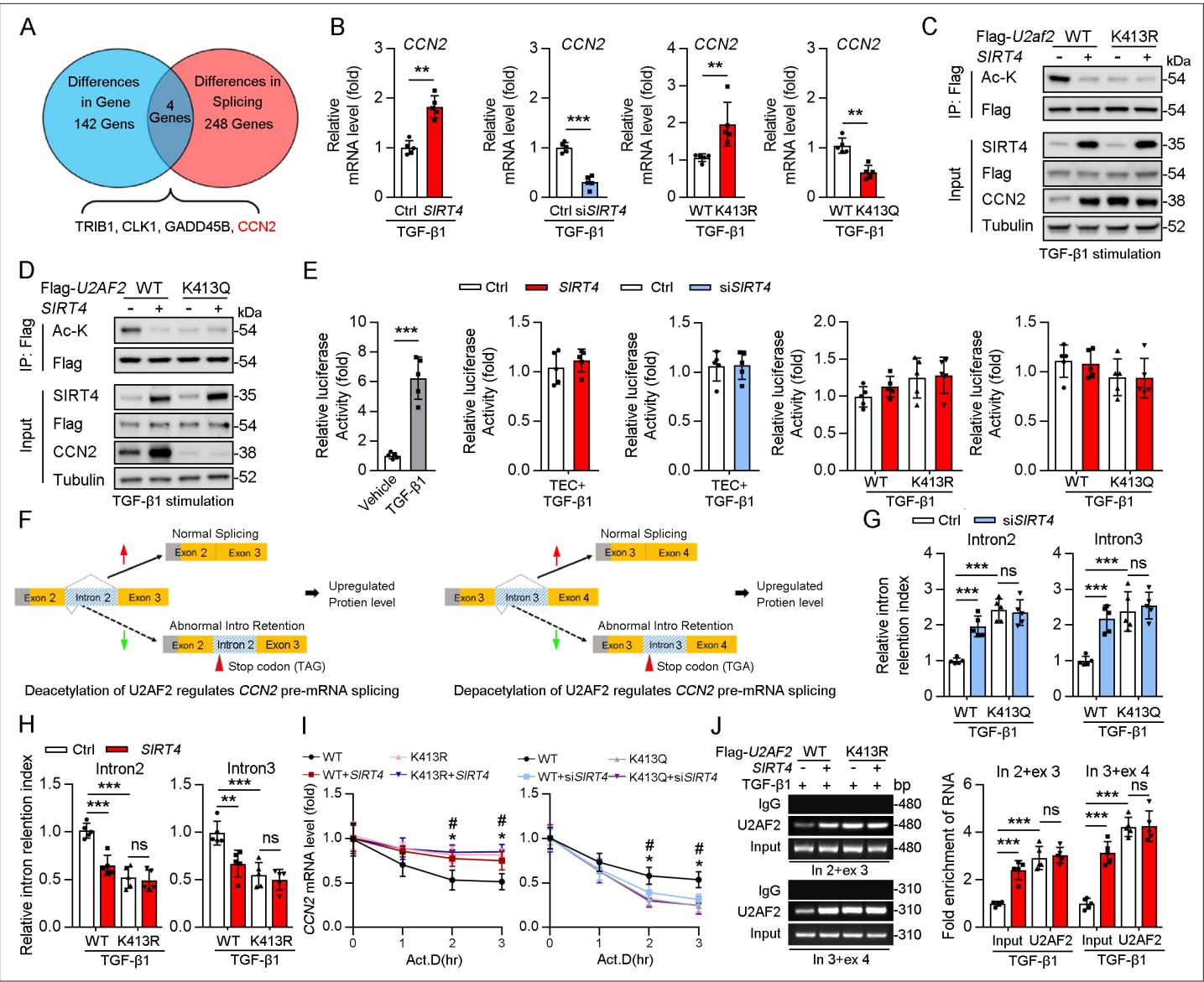

**Figure 6.** Acetylation of U2AF2 at K413 regulates CCN2 gene expression and pre-mRNA splicing. (**A**) The crosstalk of differences of the gene expression profiling analysis and splicing analysis. (**B**) The *Ccn2* mRNA level in HK2 cells (n=5). (**C, D**) HK2 cells were transfected with Flag-*U2AF2* WT, K413R, or K413Q, and Ad-*SIRT4* or *SIRT4* siRNA (si*SIRT4*) as indicated in figure. Cells lysates subjected to Co-IP with anti-Flag antibody, and western blotting using indicated antibodies. (**E**) Luciferase activity of *Ccn2* promoter in HK2 cells transfected with Ad-*SIRT4*, Ad-sh*SIRT4*, WT *U2AF2*, *U2AF2* K413R, or *U2AF2* K413Q under TGF-β1 stimulation was measured (n=5). (**F**) Schematics of *Ccn2* alternative splicing pattern regulated by hyperacetylated U2AF2. Full line and dotted line represent normal splicing and abnormal intron retention, respectively. Grey and yellow boxes represent untranslated region (UTR) and protein coding region (CDS), respectively. The first stop codon in intron 2 or 3 was indicated with red arrow. (**G, H**) Quantitative real-time PCR analysis of normal or abnormal splicing isoforms of *CCN2* (n=5). (**I**) Actinomycin D (Act D, 2 mg/mL) treatment and quantitative real-time PCR were performed to measure *CCN2* mRNA levels (n=5). (**J**) HK2 cells were transfected with Flag-*U2AF2* WT, Flag-*U2AF2* WT, or Flag-*U2AF2* K413R with or without Ad-*SIRT4* under TGF-β1 stimulation. Cells lysates subjected to RNA Binding Protein Immunoprecipitation (RIP) with anti-U2AF2 antibody (left panel). Quantitative real-time PCR results of RIP assays (right panel). For all panels, data are presented as mean ± SD. **p< 0.01, ***p< 0.001, by Student's t-test for B. **p<0.01, ***p<0.001 by one-way ANOVA with Bonferroni correction test for E, G, H, I, and J.

The online version of this article includes the following source data for figure 6:

**Source data 1.** Original files for western blot and Chip analysis displayed in *Figure 6C, D and J*.

**Source data 2.** The uncropped gels or blots with the relevant bands clearly labeled in *Figure 6C, D and J*.

splicing can trigger nonsense-mediated mRNA decay, gene expression could be affected by a change in splicing pattern. The mRNA levels of *CCN2* were remarkably increased in SIRT4[OE] cells but reduced in *Sirt4*-knockdown cells (*Figure 6B*). In addition, *CCN2* mRNA levels were significantly increased in U2AF2 K413R-transfected cells but remarkably reduced in U2AF2 K413Q (lysine to glutamine [Q] mutant for protein hyperacetylation mimic)-transfected cells (*Figure 6B*). This finding prompted us to investigate the role of specific lysine residues within U2AF2 that may be critical for the regulation of CCN2 expression. To this end, we introduced the K453Q mutation as a control to discern the distinct effects of lysine acetylation at different sites. Our results showed that SIRT4 OE significantly elevated the protein levels of CCN2 in U2AF2 WT or U2AF2 K453Q-transfected cells but not in U2AF2-K413Q or K413R-transfected cells (*Figure 6C and D*), suggesting that only the U2AF2 acetylation at K413 is efficient to regulate CCN2 expression.

CCN2 is transcriptionally regulated by Sp1 in response to TGF-β1 stimulation (*Holmes et al., 2003*). Thus, we examined whether SIRT4 induced CCN2 expression through transcription co-activator function. An analysis of the promoter region of *Ccn2* (including Sp1 binding site) showed that there was no increase in the promoter activity following overexpression of U2AF2 WT, U2AF2-K413R or U2AF2-K413Q upon TGF-β1 stimulation (*Figure 6E*), implying that the SIRT4-induced CCN2 expression is not transcription dependent. According to the RNA sequencing data, SIRT4 stimulation increased the efficiency of *Ccn2* pre-mRNA exon2/intron2 and exon3/intron3 splicing (*Figure 6F*). Upon analyzing the gene sequence of *Ccn2*, we found that both *Ccn2* introns 2 and 3 contain a TGA stop codon, implying that they could trigger the degradation of abnormal mRNAs. Furthermore, we confirmed that SIRT4 knockdown or U2AF2 K413Q increased the retention index of *Ccn2* introns 2 and 3, whereas SIRT4 OE or U2AF2 K413 reduced the retention index (*Figure 6G and H*). U2AF2 K413R or K413Q impeded the changes in intron 2 and 3 retention indices of *Ccn2* caused by SIRT4 OE or *Sirt4* knockdown, respectively (*Figure 6G and H*). Next, we assessed whether U2AF2 regulates CCN2 expression by regulating mRNA stability. We treated SIRT4 OE or *Sirt4* knockdown cells with Actinomycin D to block transcription and measured *Ccn2* mRNA stability over time. The mRNA stability of *Ccn2* was significantly enhanced in SIRT4 OE cells, whereas it was reduced in *Sirt4* knockdown cells compared to that in control cells (*Figure 6I*). As expected, compared to that in U2AF2 WT-transfected cells, U2AF2 K413R increased the mRNA stability of *Ccn2*, while U2AF2 K413Q reduced its stability (*Figure 6I*). In addition, K413R and K413Q repressed the changes in the stability of *Ccn2* mRNA caused by SIRT4 OE or *Sirt4* knockdown, respectively (*Figure 6I*). Next, a RiboIP experiment was performed wherein U2AF2 was immunoprecipitated to determine whether it can bind endogenous *Ccn2* transcripts (*Figure 6J*). SIRT4 OE enhanced the interaction between U2AF2 and *Ccn2* pre-mRNA (indicated by the PCR product containing intron 2 and exon 3 or intron 3 and exon 4) in U2AF2 WT-transfected cells but had little effect on their interaction in U2AF2 K413R-transfected cells (*Figure 6J*). This indicates that the SIRT4-induced deacetylation of U2AF2 at K413 increases CCN2 expression by regulating the alternative splicing of *Ccn2*.

## SIRT4 translocates from mitochondria to the cytoplasm through the BAX/BAK pore after TGF-β stimulation

SIRT4 is located in the mitochondrial matrix under normal conditions (*Argmann and Auwerx, 2006*). A previous study showed that SIRT4 translocates from the mitochondria into the cytoplasm upon Wnt stimulation (*Wang et al., 2022*). Hence, we conducted an immunofluorescence assay and organelle separation experiment to compare the localization of SIRT4 before and after TGF-β1 treatment. Interestingly, SIRT4 localization changed from mitochondria to the cytoplasm or even nucleus at after 12 hr of TGF-β1 treatment (*Figure 7A–C*). Moreover, TGF-β1 stimulation did not induce U2AF2 release from the nucleus into the cytoplasm, even after 24 hr (data not shown). Other stimulations such as serum starvation, tunicamycin, poly (I: C), and cisplatin also induced this translocation of SIRT4 (*Figure 7D*). It is well known that BAX and BAK are two key molecules of the mitochondrial permeability transition pore. They form polymers on the mitochondrial outer membrane and mediate the release of mitochondrial contents such as mtDNA, mitochondrial dsRNA, and cytochrome (*Ma et al., 2020*; *McArthur et al., 2018*). Therefore, we tested whether the TGF-β1-induced release of SIRT4 is dependent on the BAX/BAK oligomeric pore. Notably, BAX or BAK deficiency or MSN-125 (an effective oligomeric inhibitor of Bax and Bak) treatment almost abolished the release of SIRT4 from mitochondria to the cytoplasm as well as the upregulation of *Ccn2* under TGF-β1 stimulation (*Figure 7E and F*). As

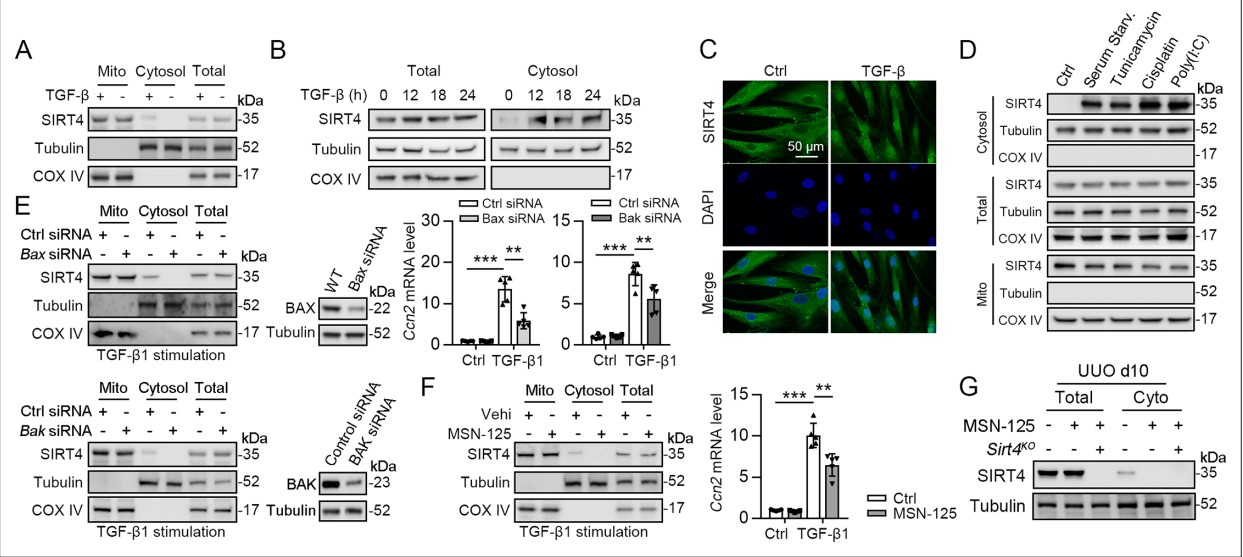

**Figure 7.** SIRT4 is released through BAX/BAK pore in a TGF-β1-dependent manner. (**A**) Organelle separation experiment and immunoblot analysis detected the expression/localization of SIRT4, Tubulin, and COX IV. (**B**) TECs were treated with TGF-β1 for indicated time. Organelle separation experiment and immunoblot analysis detected the localization of SIRT4, Tubulin, and COX IV. (**C**) Representative image of immunofluorescent staining of SIRT4 in TECs treated with TGF-β1 or not (scale bar, 50 μm). (**D**) TECs were treated with Tunicamycin, Cisplatin, Poly(I:C), or under serum starvation. Organelle separation experiment and immunoblot analysis detected the localization of SIRT4, Tubulin, and COX IV. (**E**) WT TECs incubated with *Bax* siRNA or *Bak* siRNA under TGF-β1 stimulation. Organelle separation experiment and immunoblot analysis detected the localization of SIRT4, Tubulin, and COX IV (left panels). Western blot analysis of the expression of BAX, BAK and Tubulin in TECs (middle panels). The mRNA level of *Ccn2* was detected by qPCR (n=5) (right panels). (**F**) WT TECs were treated with MSN-125 (10 μM) or vehicle. Organelle separation experiment and immunoblot analysis detected the localization of SIRT4, Tubulin, and COX IV (left panel). The mRNA level of *Ccn2* was detected by qPCR (n=5) (right panel). (**G**) Kidney tissues from WT or S4KO mice were treated with MSN-125 or vehicle. The mRNA level of *Ccn2* was detected by qPCR in the kidney of mice (n=5). For all panels, data are presented as mean ± SD. **p<0.01, ***p<0.001 by one-way ANOVA with Bonferroni correction test.

The online version of this article includes the following source data for figure 7:

**Source data 1.** Original files for western blot analysis displayed in *Figure 7A, B, D, E, F and G*.

**Source data 2.** The uncropped gels or blots with the relevant bands clearly labeled in *Figure 7A, B, D, E, F and G*.

expected, the in vivo results showed that MSN-125 inhibited the translocation of SIRT4 from mitochondria to the cytoplasm in the kidneys of mice following UUO surgery (*Figure 7G*). Together, these results suggest that TGF-β1 induces the translocation of SIRT4 from mitochondria to the cytoplasm in a BAX/BAK-dependent manner.

## ERK phosphorylates SIRT4 at Ser36 to promote the binding of SIRT4 to importin α1 and nuclear translocation

As SIRT4 showed nuclear localization (*Figure 7C*) and interacted with the nuclear protein U2AF2 (*Figure 4C*) under TGF-β1 stimulation, we investigated the mechanisms underlying SIRT4 accumulation in the nucleus. Pretreatment of TECs with LY290042 (phosphoinositide 3-kinase inhibitor), SU6656 (Src inhibitor), SP600125 (JNK inhibitor), and U0126 (MEK/ERK inhibitor) blocked TGF-β1-induced phosphorylation of AKT, c-Src, c-Jun, and ERK1/2, respectively. Immunoblotting analyses showed that only U0126 treatment abrogated the TGF-β1-induced nuclear translocation of SIRT4 (*Figure 8A*, upper panel). Compared to the amount of cytosolic SIRT4, nuclear SIRT4 is present in a small portion (*Figure 8A*, bottom panel). These results were further supported by the immunofluorescence analyses (*Figure 8B*). Additionally, expression of the Flag-ERK2 K52R kinase-dead mutant blocked the TGF-β1-induced nuclear accumulation of SIRT4 and resulted in the accumulation of phosphorylated SIRT4 in the cytosol (*Figure 8C*, left panel). Co-expression of a constitutively active MEK1 Q56P mutant (expression of constitutively active MEK1 Q56P with WT ERK2) with Flag-tagged ERK2 WT or ERK2 K52R in TECs (*Figure 8C*, right panel) showed that expression of WT ERK2, but not ERK2 K52R, induced the nuclear translocation of SIRT4. These results indicate that ERK activation is required for the TGF-β1-induced nuclear translocation of SIRT4.

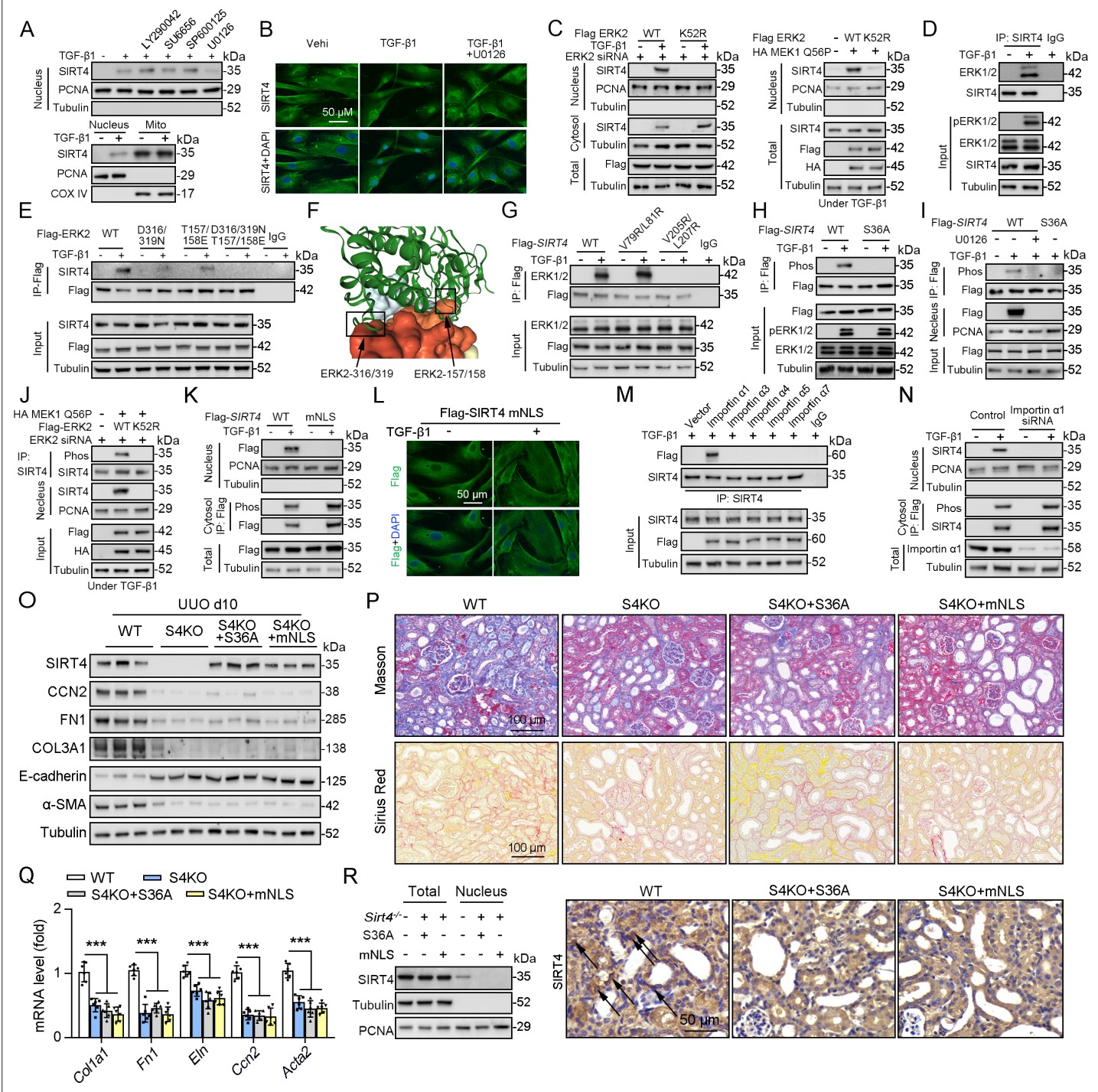

**Figure 8.** ERK2 phosphorylates SIRT4 at S36 and promotes it nucleus translocation. (**A**) (The upper panel) Nuclear fractions were prepared from TECs pretreated with LY290042 (30 μM), SU6656 (4 μM), SP600125 (25 μM), and U0126 (20 μM) for 30 min before TGF-β1 (2 ng/ml) for 12 hr. Nuclear PCNA and cytoplasmic Tubulin were used as controls. (The bottom panel) TECs were treated with TGF-β1 (2 ng/ml) for 12 hr. Organelle separation experiment and immunoblot analysis detected the localization of SIRT4, PCNA, and Tubulin. (**B**) TECs were pretreated with or without U0126 (20 μM) for 30 min, then treated with TGF-β1 (2 ng/ml) for 12 hr. Immunofluorescence analyses were performed with the indicated antibodies. (**C**) HK2 cells were stably transfected with Flag-ERK2, Flag-ERK2 K52R, or siRNA ERK2 (left panel) or transiently transfected with HA-MEK1 Q56P and indicated Flag-tagged ERK2 proteins (right panel). The cells were treated with or without TGF-β1 (2 ng/ml) for 12 hr, and the total cell lysates, cytosol and nuclear fractions were prepared for determination of indicated proteins by western blot. (**D**) TECs were treated with TGF-β1, and cells lysates subjected to Co-IP with anti-SIRT4 antibody, and western blotting using indicated antibodies. (**E**) HK2 cells transfected with vectors expressing the indicated Flag-tagged ERK proteins were treated with or without TGF-β1 (2 ng/ml) for 12 hr. (**F**) SIRT4-ERK2 docking with the HDOCK server. (**G**) HK2 cells expressing the indicated

*Figure 8 continued on next page*

*Figure 8 continued*

Flag-*SIRT4* proteins were treated with or without TGF-β1 (2 ng/ml) for 12 hr. (**H, I**) HK2 cells transfected with Flag-Sirt4 WT or S36A and then incubated with TGF-β1 and/or U0126 as indicated in figure, the total cell lysates and nuclear fractions were prepared. Total cells lysates subjected to Co-IP with anti-Flag antibody, and Western blotting using indicated antibodies. (**J**) HK2 cells were transfected with MEK1 Q56P, Flag-ERK2 WT or K52R, and ERK2 siRNA as indicated in figure, and the total cell lysates and nuclear fractions were prepared. Total cells lysates subjected to Co-IP with SIRT4 antibody and western blotting using indicated antibodies. (**K, L**) HK2 cells transfected with Flag-*Sirt4* WT or mNLS with or without TGF-β1 treatment, and the total cell lysates, cytosol, and nuclear fractions were prepared. Western blotting using indicated antibodies (**K**). Immunofluorescence analyses were performed with the indicated antibodies (**L**). (**M**) The indicated Flag-importin α proteins were transfected in HK2 cells, then treated with TGF-β1 (2 ng/ml) for 12 h. Flag-importin α proteins were immunoprecipitated with an anti-Flag antibody. (**N**) HK2 cells transfected with importin α1 were treated with or without TGF-β1 (2 ng/ml) for 12 h. The total cell lysates, cytosolic and nuclear fractions were prepared and western blotting using indicated antibodies. (**O–R**) S4KO mice received in situ renal injection of AAV9-*Ksp-Sirt4 S36A* (S36A) or AAV9-*Ksp-Sirt4 mNLS* (mNLS) at 6 weeks of age. After 2 weeks, the mice were randomly assigned to sham surgery or UUO surgery according to an established protocol. O: Western blot analysis of the expression of SIRT4, CCN2, FN1, COL3A1, E-cadherin, α-SMA and Tubulin in mouse kidney. P: Representative images of Masson's trichrome staining and Sirius red staining in kidney sections from mice (scale bar, 100 μm). Q: The mRNA level of *Col1a1, Fn1, Eln, Ccn2 and Acta2* in the kidney of mice. R: The total cell lysates and nuclear fractions were prepared from kidney and western blotting using indicated antibodies (left panel). (Right panel) Representative images of immunohistochemical staining of SIRT4 in kidneys from mice (scale bar, 100 μm). For all panels, data are presented as mean ± SD. ***p<0.001 by one-way ANOVA with Bonferroni correction test.

The online version of this article includes the following source data and figure supplement(s) for figure 8:

**Source data 1.** Original files for western blot analysis displayed in *Figure 8A, C, D, E, G, H, I, J, K, M, N, O and R*.

**Source data 2.** The uncropped gels or blots with the relevant bands clearly labeled in *Figure 8A, C, D, E, G, H, I, J, K, M, N, O and R*.

**Figure supplement 1.** U0126 prevents SIRT4 overexpression induced kidney fibrosis in UUO mice.

**Figure supplement 1—source data 1.** Original files for western blot analysis displayed in *Figure 8—figure supplement 1A*.

**Figure supplement 1—source data 2.** The uncropped gels or blots with the relevant bands clearly labeled in *Figure 8—figure supplement 1A*.

**Figure supplement 2.** Exosomes contain anti-SIRT4 antibody alleviated UUO-induced kidney fibrosis.

**Figure supplement 2—source data 1.** Original files for western blot analysis displayed in *Figure 8—figure supplement 2A, D*.

**Figure supplement 2—source data 2.** The uncropped gels or blots with the relevant bands clearly labeled in *Figure 8—figure supplement 2A, D*.

**Figure supplement 3.** Inhibition of nucleus accumulation of β-catenin failed to suppresses kidney fibrosis induced by SIRT4 overexpression in UUO mice.

**Figure supplement 3—source data 1.** Original files for western blot analysis displayed in *Figure 8—figure supplement 3A*.

**Figure supplement 3—source data 2.** The uncropped gels or blots with the relevant bands clearly labeled in *Figure 8—figure supplement 3A*.

To further determine the relationship between ERK1/2 and SIRT4, we performed a Co-IP assay and found that TGF-β1 treatment resulted in the binding of ERK1/2 to SIRT4 (*Figure 8D*). MAP kinases bind to their substrates through a docking groove comprising an acidic common docking (CD) domain and glutamic acid-aspartic acid (ED) pockets (*Lu and Xu, 2006*). Our results showed that mutation of either the ERK2 CD domain (D316/319 N) or the ED pocket (T157/158E) reduced the binding to endogenous SIRT4 compared to that in the WT ERK2 control (*Figure 8E*). Combined mutations at both the CD domain and ED pocket (T/E-D/N) abrogated the binding of ERK2 to SIRT4 entirely (*Figure 8E*), indicating that ERK2 binds SIRT4 through its docking groove. Furthermore, the server provided docking information regarding SIRT4–ERK2, indicating SIRT4 and ERK2 CD domain and ED pocket interactions (*Figure 8F*). ERK substrates often have a docking domain characterized by a cluster of basic residues, followed by an LXL motif (L represents Leu, but can also be Ile or Val; X represents any amino acid) (*Lu and Xu, 2006*). An analysis of the docking information of SIRT4 - ERK2 and the amino acid sequence of SIRT4 revealed the putative ERK-binding sequence 77-EKVGLYARTDRR-88 and 203-GDVFLSE-209, which contain LXL motifs at V79/L81 and V205/L207, respectively. Immunoblotting of the immunoprecipitated Flag-SIRT4 proteins with an anti-ERK1/2 antibody showed that a SIRT4 V205/L207 mutant, but not a SIRT4 V79/L81 mutant, markedly reduced its binding to ERK1/2 (*Figure 8G*). These results indicate that the ERK2 docking groove binds to a CD domain in SIRT4 at V205/L207. Sequence analysis of SIRT4 revealed that it contains an ERK consensus phosphorylation motif (Ser-Pro) at the S36/P37 residues. Notably, S36A mutation completely abrogated the TGF-β1-dependent phosphorylation of SIRT4 (*Figure 8H*). Consistently, pretreatment with U0126 blocked the TGF-β1-induced S36 phosphorylation and nuclear translocation of SIRT4 (*Figure 8I*). In addition, expression of constitutively active MEK1 Q56P with WT ERK2, but not of ERK2 K52R, induced SIRT4 phosphorylation (*Figure 8J*). These results indicate that ERK2 specifically phosphorylates SIRT4.

Notably, we found that SIRT4 contains a potential nuclear localization signal (NLSs), a single type containing 3–5 basic amino acids with the weak consensus Lys-Arg/Lys-X-Arg/Lys (*Miyamoto et al., 1997*), 248-KRVK-251. We mutated the K248/251 and R249 residues in the putative NLS sequences of SIRT4 to alanine (named as mNLS). Cell fractionation and immunofluorescence analyses showed that Flag-SIRT4-mNLS, unlike the WT SIRT4, was unable to translocate into the nucleus upon TGF-β1 treatment (*Figure 8K and L*). These results indicate that the NLS in SIRT4 is essential for TGF-β1-induced nuclear translocation of SIRT4. Importin α functions as an adaptor and links NLS-containing proteins to importin β, which then docks the ternary complex at the nuclear-pore complex to facilitate the translocation of these proteins across the nuclear envelope (*Cingolani et al., 2002*; *Goldfarb et al., 2004*). Six importin α family members (α1, α3, α4, α5, α6, and α7) have been identified in humans (*Goldfarb et al., 2004*). We found that the endogenous SIRT4 only binds importin α1 (*Figure 8M*). Importin a1/Rch1 is barely detectable in the glomeruli of normal SD rat kidneys but is highly expressed in tubular cells, and the importin α1/Rch1 staining is significantly enhanced in the kidneys of diabetic rats (*Köhler et al., 2001*). Depletion of importin α1 with *Rch1* (coding for importin α1) shRNA (*Figure 8N*) largely blocked the TGF-β1-induced nuclear translocation of SIRT4 and resulted in the accumulation of phosphorylated SIRT4 in the cytosol (*Figure 8N*). In vivo, U0126 restrained the *Sirt4*$^{OE}$-induced nuclear translocation of SIRT4 (*Figure 8—figure supplement 1A, B*). Furthermore, SIRT4 S36A or SIRT4 mNLS overexpression failed to accumulate SIRT4 in the nucleus and aggravated renal fibrosis in S4KO UUO mice (*Figure 8O–R*). Taken together, these results strongly suggest that TGF-β1-induced SIRT4 nuclear translocation is mediated by SIRT4 phosphorylation at S36, which is regulated by the ERK1/2 signaling pathway.

## Exosomes containing anti-SIRT4 antibodies alleviate renal fibrosis in UUO mice

SIRT4 is necessary for maintaining mitochondrial function (*Haigis et al., 2006*; *Csibi et al., 2013*). Since knockout or complete inhibition of SIRT4 may not be an advisable therapeutic strategy, we constructed exosomes containing anti-SIRT4 antibodies (αSRIT4) to treat UUO mice. Exosomes containing αSRIT4 effectively inhibited renal fibrosis in UUO mice, accompanied by a decrease in SIRT4 expression in the nucleus, with little effect on mitochondria SIRT4 content (*Figure 8—figure supplement 2A–D*).

Reportedly, the initial stage of the canonical Wnt signaling pathway in which SIRT4 translocates from mitochondria into the cytoplasm leads to β-catenin protein accumulation (*Wang et al., 2022*). To determine whether the β-catenin accumulation is involved in the SIRT4-mediated kidney fibrosis in vivo, we generated mice expressing SIRT4$^{OE}$ TECs and subjected them to UUO surgery with or without MSAB treatment (MSAB binds to β-catenin and promotes its degradation). Our results showed that SIRT4 OE remarkably aggravated renal fibrosis under MSAB treatment (*Figure 8—figure supplement 3A–C*). Moreover, downregulation of β-catenin accumulation by MSAB can inhibit renal fibrosis in WT mice, but not the SIRT4$^{OE}$ mice (*Figure 8—figure supplement 3A–C*). These findings suggest that SIRT4-mediated the pathogenesis of renal fibrosis is independent of β-catenin accumulation.

## Discussion

Renal fibrosis, especially tubulointerstitial fibrosis, is an inevitable common pathway of progressive chronic kidney disease (*Humphreys, 2018*; *Venkatachalam et al., 2015*). However, there is a lack of information regarding the pathogenesis of renal fibrosis, which hampers the development of effective therapeutics (*Liu, 2006*). Here, we demonstrate that the nuclear translocation of SIRT4 is a prime initiator of kidney fibrosis. SIRT4 significantly accumulates in the nucleus during fibrosis following obstructed nephropathy and renal ischemia reperfusion injury. Global knockout or target deletion of *Sirt4* in TECs attenuated UUO-induced kidney fibrosis, whereas TEC-specific SIRT4$^{OE}$ aggravated the fibrosis. Mechanistically, we found that TGF-β1 promoted SIRT4 release from mitochondria through the BAX/BAK pore. Furthermore, TGF-β1 activation resulted in the nuclear translocation of SIRT4, which was mediated by the ERK1/2-dependent phosphorylation of SIRT4 at S36, and consequently the binding of SIRT4 to importin α1. Nuclear SIRT4 deacetylates U2AF2 and promotes U2 snRNP formation, which promotes the *Ccn2* pre-mRNA splicing, ultimately leading to the increased CCN2 expression. In vivo, SIRT4 S36 or NLS mutants blocked SIRT4$^{OE}$-aggravated kidney fibrosis in UUO

mice, which implied that SIRT4 nuclear translocation plays a significant role in the progression of kidney fibrosis.

Acetyl-CoA is an important energy-rich metabolite for homeostasis. In normal states, abundant acetyl-CoA in the cytosol shuttles freely in the nucleus or mitochondria to modulate the acetylation of histone or non-histone proteins. A rapid reduction in total acetyl-CoA levels in renal cells is observed after TGF-β1 stimulation (*Smith et al., 2019*). Non-enzymatic acetylation levels are strongly reduced by acetyl-CoA. Under such circumstances, few exceptional hyperacetylated proteins can function as key regulators of kidney fibrosis development. In accordance with this hypothesis, we found that U2AF2 acetylation decreased and U2 snRNP formation increased after deacetylation of U2AF2 (*Figure 4H–K*). Strikingly, U2AF2-K413, the deacetylation form, promotes the pre-mRNA splicing and expression of *Ccn2* (*Figure 6C and D*), supporting the idea that acetylation of U2AF2 is responsible for fibrotic reaction under TGF-β1. Additionally, a recent study showed that U2AF2 can directly bind and stabilize circNCAPG, which participates in the nuclear translocation of ras responsive element binding protein 1, thereby activating the TGF-β pathway and promoting glioma progression (*Li et al., 2023*). Taking this into consideration, we speculate that U2AF2 may be a positive feedback regulator of TGF-β1, although further investigations are required to ascertain this.

SIRT4 is mainly located in the mitochondria and participates in various mitochondrial metabolic processes (*Csibi et al., 2013*). Some studies have revealed a potential role of SIRT4 in fibrosis (*Yin et al., 2022*; *Luo et al., 2017*). In heart, loss of SIRT4 has been found to result in the development of fibrosis. These studies indicated the protective role of SIRT4 in mitochondria. In recent research, the authors found that SIRT4 abolishment can ameliorate CCl4-induced hepatic encephalopathy phenotypes, which was mediated by downregulating and detoxifing ammonia through the urea cycle (*Hu et al., 2023*). In our study, we indicated that SIRT4 translocates from the mitochondria to the cytoplasm, a process caused by TGF-β induced mitochondrial damage (*Figure 7*). Furthermore, the cytoplasmic SIRT4 was phosphorylated by ERK2, which is a downstream of TGF-β signaling (*Figure 8A–J*). The phosphorylated SIRT4 further translocated to the nuclear for promoting CCN2 expression by regulating the alternative splicing, then accelerated the kidney fibrosis (*Figure 5*). Our study introduces a novel concept in the field, demonstrating the nuclear translocation of SIRT4 is a key initiator of kidney fibrosis. This finding diverges from previous studies that have primarily focused on SIRT4's mitochondrial roles, highlighting a new dimension of SIRT4's function in renal pathophysiology.

Some studies have shown that SIRT4 exerts a protective effect on podocytes. SIRT4 OE prevents glucose-induced podocyte apoptosis and ROS production, thereby alleviating diabetic kidney disease (DKD; *Shi et al., 2017*). Furthermore, FOXM1 transcriptionally activates SIRT4 and inhibits NF-κB signaling and the expression of the NLRP3 inflammasome to alleviate kidney injury and podocyte pyroptosis in DKD (*Xu et al., 2021*). In the present study, we suggest a profibrotic role of SIRT4 in TECs, which was contributed by upregulated expression of CCN2. Although these are complicatedly related to one another in CKD pathophysiology, injury to TECs is considered a core element that initiates progressive fibrosis (*Friedman et al., 2013*). Therefore, we suggested that SRIT4 may perform different roles in different types of cells or subcellular organelles. Moreover, further studies on the role of SIRT4 in DKD are needed to evaluate the safety of anti-SIRT4 therapy.

Overall, our study reveals that TGF-β1 activation resulted in the nuclear translocation of SIRT4, mediated by the ERK1/2-dependent phosphorylation of SIRT4 at S36, and consequently the binding of SIRT4 to importin α1. In the nucleus, SIRT4-mediated U2AF2 deacetylation at K413, a key protein for the spliceosome, acts as a responder under TGF-β1 stimulation. SIRT4 promotes CCN2 expression through alternative pre-mRNA splicing by deacetylating U2AF2, which contributes to the progression of kidney fibrosis. These findings expand the field of epigenetic regulation of fibrogenic gene expression and provide a potential therapeutic target for kidney fibrosis.

# Materials and methods
## Studies in animals

All animal care and experimental protocols for in vivo studies conformed to the Guide for the Care and Use of Laboratory Animals published by the National Institutes of Health (NIH; NIH publication no.:85–23, revised 1996). The sample size for the animal studies was calculated based on a survey of data from published research or preliminary studies. *Sirt4$^{flox/flox}$* (C57BL/6J-*Sirt4$^{em1flox}$*/Cya; Strain

ID: CKOCMP-75387-*Sirt4*-B6J-VA), Col1a2-Cre/ERT2 mice (Cat#: C001248) and Cdh16-Cre mice (Cat#: C001452) obtained from Cyagen Biosciences (Guangzhou) Inc (Guangzhou, Guangdong, China). C57BL/6 J mice and *Sirt4^{-/-}* (C57BL/6JGpt-*Sirt4^{em13Cd1976}*/Gpt; Strain ID: T011568) mice based on C57BL/6 J background were purchased from Gempharmatech Co. Ltd (Jiangsu, Nanjing, China). *U2af2^{flox/flox}* (C57BL/6J-*U2af2^{em1(flox)Smoc}*) mice based on C57BL/6 J background were purchased from Shanghai Model Organisms. These mice were maintained in SPF units of the Animal Center of Shenzhen People's Hospital with a 12 hr light cycle from 8 a.m. to 8 p.m., 23 ± 1 °C, 60–70% humidity. Mice were allowed to acclimatize to their housing environment for 7 days before the experiments. At the end of the experiments, all mice were anesthetized and euthanized in a $CO_2$ chamber, followed by the collection of kidney tissues. All animals were randomized before treatment. Mice were treated in a blinded fashion as the drugs used for treating animals were prepared by researchers who did not carry out the treatments. No mice were excluded from the statistical analysis. Studies were performed in accordance with the German Animal Welfare Act and reporting follows the ARRIVE guidelines.

## Generation of cell type–specific *Sirt4* conditional knockout mice

To generate fibroblast-specific conditional *Sirt4* knockout mouse line, *Sirt4^{flox/flox}* mice were bred with Col1a2-Cre/ERT2 transgenic mice. To activate the Cre-ERT system, tamoxifen (80 mg/kg/day, dissolved in olive oil) was injected intraperitoneally for 4 consecutive days 2 weeks before the induction of renal fibrosis in control and S4FKO mice. After UUO, uIRI, or FA, tamoxifen diet was administered until sacrifice in order to ensure the deletion of Sirt4 in newly generated myofibroblasts. To generate TECs-specific conditional Sirt4 knockout mouse line (S4TKO), *Sirt^{flox/flox}* mice were bred with Cdh16-Cre transgenic mice.

## Mouse kidneys were transfected with adeno-associated virus vector (AAV)

8-week-old mice received in situ renal injection with AAV9- empty vector (AAV9-*Ctrl*; control group), AAV9-*Ksp*-Sirt4 (*Sirt4^{OE}* group; *Ksp*, tubule specific promoter), AAV9-*Ksp*-wild type U2af2 (*wtU2af2*), and AAV9-*Ksp*-mutant U2af2 (*mU2af2*) at three independent points (10–15 μl virus per poin; virus injected dose: 2.5 E+11 v.g.) in the kidneys of mice (n=6). Adeno-associated virus type 9 constructs, including GV501 empty vector, *Sirt4*, *wtU2af2*, and *mU2af2* were provided by GeneChem Company (Shanghai, China).

## Mice kidney fibrotic models

Male C57BL/6, *Sirt4^{–/–}*, *Sirt^{flox/flox}*, and cell type–specific conditional knockout mice (~8–10 weeks old) were subjected to various kidney injury models to induce renal fibrosis. UUO was performed by permanent ligation of the right ureter with 6–0 silk. Ureter-ligated kidneys and contralateral kidneys (CLs), used as nonfibrotic controls, were collected 10 days after surgery. To establish uIRI, left renal pedicles were clamped with microaneurysm clips for 30 min followed by reperfusion. During uIRI surgery, mice were placed on a heating pad to maintain body temperature at 37 °C. Injured and contralateral kidneys were collected 1 day or 28 days after surgery for analyses. Folic acid (FA)–induced renal fibrosis was conducted by single intraperitoneal injection of 250 mg/kg folic acid (Sigma-Aldrich, 7876) dissolved in 0.3 M sodium bicarbonate, and mice were sacrificed 14 days after FA treatment. Mice injected with sodium bicarbonate served as vehicle control.

## Study approval

All animal care and experimental protocols for in vivo studies conformed to the Guide for the Care and Use of Laboratory Animals, published by the National Institutes of Health (NIH; NIH publication no.: 85–23, revised 1996), was approved by the Animal Care Committees of the First Affiliated Hospital of Southern University of Science and Technology (No. AUP-240227-LZ-0001–01), and were performed in compliance with the ARRIVE guidelines. Studies with human participants (*Supplementary file 1c*) were conducted in line with the Declaration of Helsinki. The studies were approved by Ethics Committee of the First Affiliated Hospital of Southern University of Science and Technology. The written consent obtained from patients was informed consent.

## Quantification and statistical analysis

All data were generated from at least three independent experiments. Each value was presented as the mean ± SD. All raw data were initially subjected to a normal distribution and analysis by

one-sample Kolmogorov-Smirnov (K-S) nonparametric test using SPSS 22.0 software. For animal and cellular experiments, a two-tailed unpaired student's t-test was performed to compare the two groups. One-way ANOVA followed by the Bonferroni's post-hoc test was used to compare more than two groups. To avoid bias, all statistical analyses were performed blindly. Statistical significance was indicated at *p<0.05, **p<0.01, and ***p<0.001.

## Protein extraction and western blot analysis

Cytoplasmic or nuclear extracts were prepared from cells or kidneys using a cytoplasmic and nuclear protein isolation kit (Cat# 78833, Thermo Fisher Scientific). Briefly, the tube was vortexed vigorously in the highest setting for 15 s to fully suspend the cell pellet (or tissues were homogenized in PBS). The tubes were incubated on ice for 10 min. Next, ice-cold cytoplasmic extraction reagent II solution was added to the tubes. The tube was vortexed and incubated on ice. The tube was centrifuged for 5 min at maximum speed in a microcentrifuge (~16,000 × $g$), and the supernatant (cytoplasmic extract) was transferred to a tube. The insoluble (pellet) fraction, which contains nuclei, was suspended in ice-cold nuclear extraction reagent and then vortexed at the highest setting for 15 s. The sample was placed on ice and vortexed for 15 s every 10 min for 40 min. The tube was centrifuged at maximum speed (~16,000 × $g$) in a microcentrifuge for 10 min, and the supernatant (nuclear extract) fraction was transferred to a tube.

Mitochondria were extracted using a Mitochondria Isolation Kit (Sigma) following the manufacturer's protocol. In brief, $2 \times 10^7$ cells were pelleted by centrifuging the harvested cell suspension, and then mitochondria isolation reagent was added to the cell pellets. The cell resuspension was centrifuged at 700 × $g$ for 10 min at 4 °C, and then the supernatant was transferred to a new 1.5 ml tube and centrifuged at 12,000 × $g$ for 15 min at 4 °C. The supernatant (cytosolic fraction) was transferred to a new tube, and the pellet contained the isolated mitochondria. The pellet was further lysed to yield the final mitochondrial lysate. The extracted proteins were prepared for subsequent western blotting analysis.

For western blot analysis, 50 µg of lysate was loaded onto sodium dodecyl sulfate-polyacrylamide gel electrophoresis gels, and transferred onto polyvinylidene difluoride membranes (Millipore). Proteins were analyzed with their corresponding specific antibodies. Densitometry analysis was performed using Quantity One Software and quantified relative to the loading control, Tubulin.

## Antibodies used in western blottings

| Antibodies | Source | Identifier |
|---|---|---|
| SIRT4 (1:1000) | Thermo Fisher Scientific | Cat# PA5-114377; RRID:AB_2787525 |
| SF3B1 (1:1000) | Thermo Fisher Scientific | Cat# MA5-42938; RRID:AB_2912079 |
| KAT5 (1:1000) | Thermo Fisher Scientific | Cat# PA5-34548; RRID:AB_2551900 |
| SF3B5 (1:1000) | Thermo Fisher Scientific | Cat# A305-586A-T; RRID:AB_2782745 |
| SF3B3 (1:1000) | Abcam | Cat# ab209402; RRID:AB_2910146 |
| CCN2 (1:1000) | Abcam | Cat# ab6992; RRID:AB_305688 |
| FN1 (1:1000) | Abcam | Cat# ab2413; RRID:AB_2262874 |
| α-SMA (1:1000) | Abcam | Cat# ab7817; RRID:AB_262054 |
| U2AF2 (1:50/1:250) | Abcam | Cat# ab37530; RRID:AB_883336 |
| Flag (1:50/1:1000) | Abcam | Cat# ab205606; RRID:AB_2916341 |
| Anti-acetyl Lysine (1:1000) | Abcam | Cat# ab21623; RRID:AB_446436 |
| PCAF (1:1000) | Abcam | Cat# ab96510; RRID:AB_10679933 |
| KAT8 (1:1000) | Abcam | Cat# ab200660; RRID:AB_2891127 |
| COX IV (1:1000) | Abcam | Cat# ab202554; RRID:AB_2861351 |
| KPNA2 (1:1000) | Abcam | Cat# ab84440; RRID:AB_1860701 |
| p300 (1:1000) | Abcam | Cat# ab275378; RRID:AB_2935873 |
| HAT1 (1:1000) | Abcam | Cat# ab194296; RRID:AB_2801641 |

*Continued on next page*

*Continued*

| Antibodies | Source | Identifier |
|---|---|---|
| BAK | Abcam | Cat# ab104124; RRID:AB_10712355 |
| COL1A1 (1:1000) | Proteintech | Cat# 67288–1-Ig; RRID:AB_2882554 |
| HA-Tag (1:50/1:1000) | Cell signaling Technology | Cat# 3724; RRID:AB_1549585 |
| Poly/Mono-ADPRibose (1:1000) | Cell signaling Technology | Cat# 83732; RRID:AB_2749858 |
| CBP (1:1000) | Cell Signaling Technology | Cat# 7389; RRID:AB_2616020 |
| E-cadherin (1:1000) | Proteintech | Cat# 20874–1-AP; RRID:AB_10697811 |
| Tubulin (1:1000) | Proteintech | Cat# 11224–1-AP; RRID:AB_2210206 |
| PCNA (1:2000) | Novus | Cat# NB500-106; RRID:AB_2252058 |
| PUF60 (1:1000) | Novus | Cat# NBP1-49906; RRID:AB_10012096 |
| SF3B4 (1:1000) | Novus | Cat# NBP1-92692; RRID:AB_11025842 |
| COL3A1 (1:1000) | Novus | Cat# NB600-594; RRID:AB_10001330 |
| SF3B2 (1:1000) | Novus | Cat# NBP1-92380; RRID:AB_11031649 |
| BAX (1:1000) | Novus | Cat# NBP1-28566; RRID:AB_1852819 |
| ERK1/ERK2 (1:1000) | Novus | Cat# AF1576; RRID:AB_354872 |
| p-ERK1/2 (1:1000) | Novus | Cat# AF1018; RRID:AB_354539 |
| U2AF1 (1:100/1:1000) | Novus | Cat# NBP1-32515; RRID:AB_2288123 |

## Histological analysis

Kidney tissue was embedded in paraffin and sliced into 5-µm-thick serial sections using a paraffin slicer. For kidney histology, the paraffin sections were stained using Masson's trichrome and Sirius red staining kit (Solarbio & Technology, Beijing, China).

## Quantitative real-time PCR

Total RNAs were extracted by Trizol (Invitrogen) and then dissolved in an appropriate amount of RNase water. cDNA was obtained by a reverse transcription kit purchased from TransGen Biotech (Beijing, China). qPCR was performed using the ABI StepOnePlusTM Real-time PCR system (Applied Biosystem) with specific primers (*Supplementary file 1d*). The relative mRNA levels of target genes were analyzed using the $2^{-\Delta\Delta CT}$ method. Tubulin was used as a housekeeping gene for analysis.

## RIME

The procedure was performed as previously described. The hTECs ($2\times10^7$) were treated with DMSO or DMOG (1 mM) for 8 hr and cross-linked with 1% formaldehyde for 8 min and quenched by 0.125 M glycine. Cells were harvested and the nuclear fraction was extracted. After sonication, the cell lysate was immunoprecipitated with bead-prebound SIRT4 antibody (NB100-479, Novus Biologicals). Precipitated proteins in 30 µl of 100 mM ammonium hydrogen carbonate were reduced in 2.5 mM dithiothreitol at 60 °C for 1 hr, then alkylated with 5 mM iodoacetamide in the dark at room temperature for 15 min. Proteins were digested with 20 ng/µl Trypsin/LysC (Promega) at 37 °C overnight. Peptides were desalted on Oasis HLB µ-elution plates (Waters), eluted with 65% aceto-nitrile/0.1% trifluoroacetic acid, dried by vacuum centrifugation, and analyzed by liquid chromatog-raphy/tandem mass spectrometry using a nano-Easy LC 1000 interfaced with an Orbitrap Fusion Lumos Tribrid Mass Spectrometer (Thermo Fisher Scientific). Tandem mass spectra were extracted by Proteome Discoverer version 2.3 (Thermo Fisher Scientific) and searched against the SwissProt_Full_Synchronized_2018_08 database using Mascot version 2.6.2 (Matrix Science). Mascot '.dat' files were compiled in Scaffold version 3 (Proteome Software) to validate MS/MS-based peptide and protein identifications. Peptide identifications were accepted if false discovery rate (FDR) was less than 1%, based on a concatenated decoy database search by the Peptide Prophet algorithm with Scaffold delta-mass correction.

## Co-immunoprecipitation (co-IP)

After treatment, the cells were lysed in an ice-cold co-IP buffer containing 20 mM Tris-HCl (pH 8.0), 100 mM NaCl, 1 mM EDTA, and 0.5% NP-40, supplemented with a protease inhibitor cocktail (Roche, 04693132001), for 30 min. The cell homogenates were then centrifuged at 13,000 × $g$ for 15 min, and the resulting supernatant was incubated overnight at 4 °C on a shaker with primary antibodies (anti-Flag, anti- SIRT4, anti-HA, anti-U2AF2 or anti-IgG). To ensure complete saturation of the primary antibodies, sufficient cell lysates were cultured and collected for IP. The mixture of antibodies and proteins was subsequently incubated with protein A/G-agarose beads (Thermo Fisher Scientific, Cat#: 78610) at 4 °C for 3 hr. The beads were washed five to six times with cold IP buffer and resuspended in loading buffer. The cell lysates and immunoprecipitates were denatured in loading buffer at 95 °C for 5 min, and western blotting analysis was performed.

## Protein-protein docking simulation

Visual protein-protein docking of SIRT4 and U2AF2 or ERK2 was performed online using HDOCK (http://hdock.phys.hust.edu.cn/).

## Analysis of the conservation of lysine 413 in U2AF2

A gene panel of the NCBI database (https://www.ncbi.nlm.nih.gov/gene/) was used to download amino acid sequences of proteins from multiple species and subsequently compare sequence conservation between species using the UGENE software (version 39; the software can be downloaded from the following website: http://ugene.net/).

## Bioinformatics methods predict acetylase of Lys413 in U2AF2

A Group-based Prediction System (GPS) online tool (http://pail.biocuckoo.org) was used to predict Lys413 in U2AF2 modified by histone acetyltransferases, with high score values suggesting better results. The amino acid sequence of the protein in FASTA format was downloaded from UniProt and uploaded to the GPS website.

## Luciferase report assay for TGF-β1

Specific primers for the 5'-ACTCTCGAGCAGTGTTCCCACCCTGACAC-3' and for 5'-ACTAAGCTTGGTTGGCACTGCGGGCGGAG-3' were designed to amplify the DNA (–835/+214) in the CTGF promoter region of the human genome through the polymerase chain reaction (PCR) experiment. The sequence of the human CTGF promoter was cloned into the pGL3.0-Basic vector (LMAI Bio; Cat# LM-1554) through molecular cloning. Positive plasmids were screened and identified by methods such as monoclonal colony PCR, plasmid double enzyme digestion identification, and sequencing. Then, using Lipofectamine 3000 transfection reagent, pGL3.0-Basic-CTGF and Renilla luciferase were transfected into HEK293T. Firefly and Renilla luciferase activities were then measured by a dual luciferase reporter gene system (Promega, Madison, WI, USA).

## Cell culture, transfection

The human proximal tubular epithelial cell line (HK2 cells; Cat# CRL-2190) were purchased from the American Type Culture Collection (Manassas, VA, USA). Cells were cultured in Minimum Essential Medium (Thermo Fisher Scientific, Shanghai, China; Cat# 10373017). Mouse podocytes (MPC; Cat# BNCC342021) and mouse renal glomerular endothelial cells (GEC; Cat# BNCC360313) were purchased from BeNa Culture Collection (Beijing, China), cultured in DMEM-H complete medium (BeNa Culture Collection, Cat# BNCC338068). Mouse renal fibroblasts (MF; Procell Life Science&Technology, Wuhan, china; Cat# CP-M069) were cultured in mouse kidney fibroblast complete culture medium (Procell, Cat#CM-M069). All kinds of media were supplemented with 10% FBS (Gibco, Grand Island, NY), 100 U·mL−1 of penicillin (Gibco), and 100 mg•mL−1 of streptomycin (Gibco). All cells were cultured in a 37 °C incubator containing 5% CO2 and 95% air. Human embryonic kidney 293 cells (HEK293) were purchased from ATCC and maintained in HyClone Dulbecco's Modified Eagle Medium (SH30022, Cytiva) with 10% FBS, 1% glutamine, and 1% penicillin/streptomycin solution in a humidified incubator supplemented with 95% air/5% $CO_2$ at 37 °C. Cells were regularly checked for mycoplasma in a standardized manner, by a qPCR test, performed under ISO17025 accreditation to ensure work was conducted in mycoplasma-negative cells.

Adenovirus particles used in the article were provided by GeneChem Company (Shanghai, China). Adenovirus was used to transfect HK2 cells or TECs for 6 hr (adenovirus particles containing empty vectors were used as controls), and then replace the fresh medium and continue to culture. The commercial siRNA Ctrl (Cat# sc-37007, Santa Cruz), siRNA *Hat1* (Cat# sc-145898, Santa Cruz), siRNA *Sirt4* (Cat# sc-63025, Santa Cruz), siRNA *Bax* (Cat# sc-29213, Santa Cruz), and siRNA *Bak* (Cat# sc-29785, Santa Cruz) were transfected using Lipofectamine 3000 (Invitrogen, Carlsbad, California, USA) according to the manufacturer's instructions.

## Isolation of primary renal tubular epithelial cells (TECs)

Primary mouse renal tubular epithelial cells were isolated using an established protocol we previously published (*Yang et al., 2023*). Briefly, the cortex of the kidneys was carefully dissected and chopped into small pieces. Then, 1 mg/ml of collagenase solution was applied and incubated at 37 °C for 30 min with gentle agitation. The digestion was terminated by FBS and then filtered sequentially. Fragmented tubules were collected and maintained in a renal epithelial cell basal medium using a growth kit. The medium was changed for the first time, after 72 hr. The purity of the primary mouse renal tubular epithelial cells was confirmed by immunofluorescence staining of SGLT1 (Novus, NBP2-20338). Cells at passages 2–5 were used for the experiments.

## Immunofluorescent staining of cells

After treatment, cells was fixed for 15 min using fresh, methanol-free 4% formaldehyde, and then, rinsed thrice with PBS for 5 min each. After blocking with goat serum for 30 min, the cells were incubated with primary antibody against SIRT4 (Thermo Fisher Scientific, Cat# PA5-114377) at 4 °C overnight. Alexa Fluor488 goat antibodies against murine IgG (Invitrogen, Shanghai, China; Cat# A-11078, 1:400) were included as secondary antibodies. As negative controls, the primary antibodies were exchanged for nonimmune serum from the same species. The samples were counterstained with DAPI for 15 min. The sections were sealed with a cover glass, and the specimens were examined using the appropriate excitation wavelength. Images were captured and processed with a Laica microscope (Wetzlar, Germany).

## Immunohistochemical (IHC) staining

Paraffin-embedded mouse kidney samples were sliced into 4 µm-thick sections and subjected to IHC staining using a Rabbit two-step detection kit (Rabbit enhanced polymer detection system, ZSGB Bio, Beijing, China). Briefly, mouse kidney sections were deparaffinized, followed by antigen retrieval, treatment with peroxidase block, and incubation with rabbit anti-SIRT4 (1:100, Thermo Fisher Scientific, Cat# PA5-114377) primary antibody overnight at 4 °C. Tissue sections were then washed and incubated with the reaction-enhancing solution at room temperature for 20 min. After washing three times with phosphate buffer saline (PBS), tissue sections were treated with enhanced enzyme-labeled goat anti-rabbit IgG polymerase for 20 min and then developed with 3,3' Diaminobenzidine solution, counterstained with hematoxylin, and mounted with mounting medium. The kidney frozen slices from patients with minimal change disease and DKD were fixed with 4% paraformaldehyde for 20 min, incubated with anti-SIRT4 primary antibody overnight at 4 °C, and subjected to immunohistochemistry staining as described above. The stained area was measured using ImageJ software.

## RNA immunoprecipitation (RIP)

RIP is a powerful method to investigate the interactions between RNA and RNA binding proteins in vivo by immunoprecipitating RNA-protein complexes using specific antibodies against RNA binding proteins. An example of the RIP protocol is as follows: Covalently conjugate protein-specific antibody to the beads. Wash extensively with lysis buffer. Collect cell lysate and incubate with antibody conjugated beads allowing RNA-protein complexes formation. Use normal IgG as negative control. Incubate the lysate for 2–4 hr at 4 °C with gentle rotation. Wash beads stringently with ice-cold lysis buffer containing RNase inhibitors, protease inhibitors and nonionic detergents. Perform 4–6 quick washes. Elute RNA-protein complexes from beads, generally with 1% SDS buffer heated to 95 °C for 5 min. Extract RNA with phenol/chloroform. Isolate RNA for downstream applications like RT-qPCR or RNA sequencing.

## The stability and intron retention index of CCN2 mRNA

To detect the CCN2 mRNA stability, the cells were treated with Actinomycin D (2 mg/mL) for 0, 1, 2, and 3 hr. The cDNA from different time point was diluted in nuclease-free water for the same times, and mRNA was quantified by qPCR. The fold of changing CCN2 level was normalized to the initial expression level of CCN2 (0 hr). Intron Retention index = (mRNA level of normal splicing isoform/total CCN2 mRNA level)/ (mRNA level of abnormal splicing isoform/total CCN2 mRNA level). The CCN2 mRNA stability qPCR primers are as listed in *Supplementary file 1e*.

## Loading of exosomes

The approaches for αSIRT4 incorporation into exosomes were executed as mentioned before (*Haney et al., 2015*). Naive exosomes released from HEK293T were diluted in PBS to a concentration 0.15 mg/mL of total protein, then αSIRT4 solution in PBS (0.5 mg/mL) was added to 250 µl of exosomes to the final concentration 0.1 mg/mL total protein, and incubated at RT for 18 hr. In case of a saponin treatment, a mixture of αSIRT4 and exosomes was supplemented with 0.2% saponin and placed on shaker for 20 min at room temperature.

## Acknowledgements

This work was supported by grants from the National Natural Science Foundation of China (82370876 to Shu Yang, 82170842 and 82371572 to Zhen Liang, 82171556 to Lin Kang), Key Program Topics of Shenzhen Basic Research, China (No. JCYJ20220818102605013 to Lin Kang). Sequencing service was provided by Bioyi Biotechnology Co., Ltd. Wuhan, China.

## Additional information

### Funding

| Funder | Grant reference number | Author |
|---|---|---|
| National Natural Science Foundation of China | 82370876 | Shu Yang |
| National Natural Science Foundation of China | 82170842 | Zhen Liang |
| National Natural Science Foundation of China | 82371572 | Zhen Liang |
| National Natural Science Foundation of China | 82171556 | Lin Kang |
| Key Program Topics of Shenzhen Basic Research | JCYJ20220818102605013 | Lin Kang |

The funders had no role in study design, data collection and interpretation, or the decision to submit the work for publication.

### Author contributions

Guangyan Yang, Data curation, Writing – original draft; Jiaqing Xiang, Data curation, Investigation; Xiaoxiao Yang, Writing – review and editing; Xiaomai Liu, Data curation, Writing – review and editing; Yanchun Li, Lixing Li, Data curation; Lin Kang, Zhen Liang, Funding acquisition, Project administration; Shu Yang, Conceptualization, Writing – original draft, Project administration, Writing – review and editing

### Author ORCIDs

Shu Yang ⓘ https://orcid.org/0000-0002-4912-777X

Reviewer #1 (Public review): https://doi.org/10.7554/eLife.98524.3.sa1
Reviewer #2 (Public review): https://doi.org/10.7554/eLife.98524.3.sa2
Reviewer #3 (Public review): https://doi.org/10.7554/eLife.98524.3.sa3

Author response https://doi.org/10.7554/eLife.98524.3.sa4

## Additional files

### Supplementary files

• Supplementary file 1. Supplementary tables 1a-e. (a) Differential expressed genes in SIRT4 OE TECs vs Ctrl TECs. Differential expression was performed with DESeq, *P*<0.01. (b) Genes with significantly different intron retention events in SIRT4 OE TECs vs Ctrl TECs (FDR <0.05). (c) Clinical characteristics of CKD in biopsy samples Data are calculated by One-way ANOVA and presented as mean ± SEM. The number of patients in each group was as indicated. NS: not significant difference, y: year. (d) Primers used in qPCR. (e) Primers used in IR Index.

• MDAR checklist

### Data availability

Raw data as fastq files were deposited at Gene Expression Omnibus GSE279225.

The following dataset was generated:

| Author(s) | Year | Dataset title | Dataset URL | Database and Identifier |
|---|---|---|---|---|
| Yang G, Xiang J, Yang X, Kang L, Liang Z, Yang S | 2024 | Nuclear Translocation of SIRT4 Mediates Deacetylation of U2AF2 to Modulate Renal Fibrosis Through Alternative Splicing-mediated Upregulation of CCN2 | https://www.ncbi.nlm.nih.gov/geo/query/acc.cgi?acc=GSE279225 | NCBI Gene Expression Omnibus, GSE279225 |

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
