## [Editor Report · eLife Assessment]

This study demonstrates a novel role for SIRT4; a mitochondrial deacetylase, shown to translocate into nuclei where it regulates RNA alternative splicing by modulating U2AF2 and the gene expression of CCN2 in tubular cells in response to TGF-β. This **fundamental** work substantially advances our understanding of kidney fibrosis development and offers a potential therapeutic approach. The evidence supporting the conclusions of a SIRT4-U2AF2-CCN2 axis activated by TGF-β is **compelling** and adds a new layer of complexity to the pathogenesis of chronic kidney disease.

---

## [Referee Report · Reviewer #1 (Public review)]

In this manuscript, Yang et al report a novel regulatory role of SIRT4 in the progression of kidney fibrosis. The authors showed that in the fibrotic kidney, SIRT4 exhibited an increased nuclear localization. Deletion of Sirt4 in renal tubule epithelium attenuated the extent of kidney fibrosis following injury, while overexpression of SIRT4 aggravates kidney fibrosis. Employing a battery of in vitro and in vivo experiments, the authors demonstrated that SIRT4 interacts with U2AF2 in the nucleus upon TGF-β1 stimulation or kidney injury and deacetylates U2AF2 at K413, resulting in elevated CCN2 expression through alternative splicing of Ccn2 gene to promote kidney fibrosis. The authors further showed that the translocation of SIRT4 is through the BAX/BAK pore complex and is dependent on the ERK1/2-mediated phosphorylation of SIRT4 at S36, and consequently the binding of SIRT4 to importin α1. This fundamental work substantially advances our understanding of the progression of kidney fibrosis and uncovers a novel SIRT4-U2AF2-CCN2 axis as a potential therapeutic target for kidney fibrosis.

Comment on revised version:

In the new version of the manuscript, the authors have addressed most of my concerns . Overall, the authors have done an extensive, well-performed study. The results are convincing, and the conclusions are mostly well supported by the data. The message is interesting to a wider community working on kidney fibrosis, protein acetylation and SIRT4 biology. This work substantially advances our understanding of the mechanism of kidney fibrosis and uncovers a novel SIRT4-U2AF2-CCN2 axis as a potential therapeutic target for kidney fibrosis.

---

## [Referee Report · Reviewer #2 (Public review)]

Summary:

The manuscript by Yang et al. presents a novel and significant investigation into the role of SIRT4 For CCN2 expression in response to TGF-β by modulating U2AF2-mediated alternative splicing and its impact on the development of kidney fibrosis.

Strengths:

The authors' main conclusion is that SIRT4 plays a role in kidney fibrosis by regulating CCN2 expression via pre-mRNA splicing. Additionally, the study reveals that SIRT4 translocates from the mitochondria to the cytoplasm through the BAX/BAK pore under TGF-β stimulation. In the cytoplasm, TGF-β activated the ERK pathway and induced the phosphorylation of SIRT4 at Ser36, further promoting its interaction with importin α1 and subsequent nuclear translocation. In the nucleus, SIRT4 was found to deacetylate U2AF2 at K413, facilitating the splicing of CCN2 pre-mRNA to promote CCN2 protein expression. Overall, the findings are fully convincing. The current study, to some extent, shows potential importance in this field.

---

## [Referee Report · Reviewer #3 (Public review)]

Summary:

Yang et al reported in this paper that TGF-beta induces SIRT4 activation, TGF-beta activated SIRT4 then modulates U2AF2 alternative splicing, U2AF2 in turn causes CCN2 for expression. The mechanism is described as this: mitochondrial SIRT4 transport into the cytoplasm in response to TGF-β stimulation, phosphorylated by ERK in the cytoplasm, and pathway and then undergo nuclear translocation by forming the complex with importin α1. In the nucleus, SIRT4 can then deacetylate U2AF2 at K413 to facilitate the splicing of CCN2 pre-mRNA to promote CCN2 protein expression. Moreover, they used exosomes to deliver Sirt4 antibodies to mitigate renal fibrosis in a mouse model. TGF-beta has been widely reported for its role in fibrosis induction.

Strengths:

TGF-beta induction of SIRT4 translocation from mitochondria to nuclei for epigenetics or gene regulation remains largely unknown. The findings presented here that SIRT4 is involved in U2AF2 deacetylation and CCN2 expression are interesting.

Comments on revised version:

I went through the revised manuscript and the letter from the authors. I have no further concerns.

---

## [Author Response]

The following is the authors’ response to the original reviews.

**Public Reviews:**

**Reviewer #1 (Public Review):**
Summary:In this manuscript, Yang et al report a novel regulatory role of SIRT4 in the progression of kidney fibrosis. The authors showed that in the fibrotic kidney, SIRT4 exhibited an increased nuclear localization. Deletion of Sirt4 in renal tubule epithelium attenuated the extent of kidney fibrosis following injury, while overexpression of SIRT4 aggravates kidney fibrosis. Employing a battery of in vitro and in vivo experiments, the authors demonstrated that SIRT4 interacts with U2AF2 in the nucleus upon TGF-β1 stimulation or kidney injury and deacetylates U2AF2 at K413, resulting in elevated CCN2 expression through alternative splicing of Ccn2 gene to promote kidney fibrosis. The authors further showed that the translocation of SIRT4 is through the BAX/BAK pore complex and is dependent on the ERK1/2-mediated phosphorylation of SIRT4 at S36, and consequently the binding of SIRT4 to importin α1. This fundamental work substantially advances our understanding of the progression of kidney fibrosis and uncovers a novel SIRT4-U2AF2-CCN2 axis as a potential therapeutic target for kidney fibrosis.Strengths:Overall, this is an extensive, well-performed study. The results are convincing, and the conclusions are mostly well supported by the data. The message is interesting to a wider community working on kidney fibrosis, protein acetylation, and SIRT4 biology.Weaknesses:The manuscript could be further strengthened if the authors could address a few points listed below:(1) In the results part 3.9, an in vitro deacetylation assay employing recombinant SIRT4 and U2AF2 should be included to support the conclusion that SIRT4 is a deacetylase of U2AF2. Similarly, an in vitro binding assay can be included to confirm whether SIRT4 and U2AF2 are directly interacted.

Thank you for your insightful comments and suggestions for improving our manuscript. We appreciate your recommendation to include an in vitro deacetylation assay employing recombinant SIRT4 and U2AF2 to support our conclusion regarding the deacetylase activity of SIRT4 on U2AF2.

We would like to clarify that the data demonstrating the effect of SIRT4 on U2AF2 acetylation were already included in our original submission. Specifically, Figure 5C illustrates that the TGF-β1-caused decreased acetylation of U2AF2 is attenuated by Sirt4 knockdown. Conversely, overexpression of SIRT4 (SIRT4 OE) enhances the deacetylation process of U2AF2 in the presence of TGF-β1. These results support that SIRT4 is a deacetylase for U2AF2.

Furthermore, we have already provided evidence of the direct interaction between SIRT4 and U2AF2 through a co-immunoprecipitation (CoIP) assay, which was shown in Figure 5B. This assay confirms the physical interaction between SIRT4 and U2AF2.

We believe that the existing data sufficiently address the points raised in your comments. We are grateful for the opportunity to clarify these aspects of our study and hope that our response has adequately addressed your concerns.

(2) In Figure 6D, the Western Blot data using U2AF2-K453Q is confusing and is quite disconnected from the rest of the data and not explained. This data can be removed or explained why U2AF2-K453Q is employed here.

Thank you for your inquiry regarding the rationale behind the K453Q mutation in our study.

In the study, we have predicted some acetylation sites. U2AF2-K453Q is another site mutation to mimic a hyperacetylated state of U2AF2, our results indicated that U2AF2 acetylation at K413 had little effects on CNN expression. Therefore, we found that only the U2AF2 acetylation at K413 can regulate CCN2 expression, not acetylation at other sites. In order not to cause ambiguity in the study, we have removed the results of U2AF2-K453Q in our revised manuscript.

(3) Although ERK inhibitor U0126 blocked the nuclear translocation of SIRT4 in vivo, have the authors checked whether treatment with U0126 could affect the expression of kidney fibrosis markers in UUO mice?

Thank you for your insightful question regarding the effects of the ERK inhibitor U0126 on the expression of kidney fibrosis markers in UUO mice.

In our study, we indeed conducted in vivo experiments using U0126 and observed that it effectively ameliorated kidney fibrosis markers, which is consistent with its established role in inhibiting the fibrotic process. Specifically, U0126 treatment significantly suppressed the SIRT4-mediated renal fibrosis, which was evidenced by the reduced expression of fibrosis markers (Author response image 1).

**Author response image 1. sa4fig1:** U0126 treatment alleviates renal fibrosis in UUO mice.

However, in the initial submission, we chose not to include these results in the main body of the manuscript based on the following reasons: (1) we intent to highlight the inhibitory effects of U0126 on ERK and its subsequent impact on kidney fibrosis might shift the focus of our study away from the central theme of SIRT4's role in renal fibrosis. (2) We aimed to maintain a clear narrative that emphasizes the novel findings related to SIRT4 and its regulation by the ERK pathway.

Nonetheless, we recognize the importance of these findings and are willing to include the relevant data in the revised manuscript if it aligns with the journal's editorial direction and contributes to the broader understanding of renal fibrosis treatment strategies.

We appreciate the opportunity to clarify this aspect of our research and are open to further suggestions from the editorial team.

(4) The format of gene and protein abbreviations in the manuscript should be standardized.

Thank you for your comment on the formatting of gene and protein abbreviations in our manuscript. We have carefully reviewed our formatting practices and confirmed that we have adhered to the standard conventions as follows:

(1) Mouse gene names are presented with an initial capital letter and in italics.

(2) Human gene names are written in uppercase and in italics.

(3) Protein names are in all capital letters and not italicized.

We understand the importance of consistency in scientific publications and have ensured that these standards are uniformly applied throughout the revised manuscript. If there were any discrepancies, we have corrected them to maintain the clarity and professionalism.

We appreciate the opportunity to refine our work and are committed to upholding the standards of scientific communication.

(5) There are a few grammar issues throughout the manuscript. The English/grammar could be stronger, thus improving the overall accessibility of the science to readers.

Thank you for bringing the grammar issues to our attention. We have made diligent efforts to revise and improve the manuscript's English and grammar throughout. We have also enlisted the support of a professional language editing service to ensure the clarity and accuracy of our scientific communication.

We are confident that these revisions have significantly enhanced the manuscript's accessibility to a broader readership and have addressed the language concerns raised.

We appreciate your guidance and are committed to delivering a manuscript of the highest quality.

**Reviewer #2 (Public Review):**
Summary:This manuscript presents a novel and significant investigation into the role of SIRT4 For CCN2 expression in response to TGF-β by modulating U2AF2-mediated alternative splicing and its impact on the development of kidney fibrosis.Strengths:The authors' main conclusion is that SIRT4 plays a role in kidney fibrosis by regulating CCN2 expression via pre-mRNA splicing. Additionally, the study reveals that SIRT4 translocates from the mitochondria to the cytoplasm through the BAX/BAK pore under TGF-β stimulation. In the cytoplasm, TGF-β activated the ERK pathway and induced the phosphorylation of SIRT4 at Ser36, further promoting its interaction with importin α1 and subsequent nuclear translocation. In the nucleus, SIRT4 was found to deacetylate U2AF2 at K413, facilitating the splicing of CCN2 pre-mRNA to promote CCN2 protein expression. Overall, the findings are fully convincing. The current study, to some extent, shows potential importance in this field.Weaknesses:(1) Exosomes containing anti-SIRT4 antibodies were found to effectively mitigate UUO-induced kidney fibrosis in mice. While the protein loading capacity and loading methods were not mentioned.

We appreciate your inquiry about the protein loading capacity and methods for the exosomes. As you have correctly noted, these details are indeed essential for the comprehensive understanding of our experimental approach. We have provided these information in the electronic supplementary material, specifically in Section 2.17, where we describe the methodology used for loading the anti-SIRT4 antibodies into the exosomes and the capacity at which this was achieved.

We hope that this additional detail in the supplementary material addresses your concerns and enhances the clarity of our study's methodology.

(2) The method section is incomplete, and many methods like cell culture, cell transfection, gene expression profiling analysis, and splicing analysis, were not introduced in detail.

Thank you for your meticulous review and the feedback provided on our manuscript. We acknowledge your concern regarding the completeness of the methods section.

We would like to clarify that in our initial submission, all text and figures were compiled into a single document, with the supplementary methods detailed at the end, separate from the main text methods. This format was chosen to adhere to submission guidelines that prioritize the concise presentation of core methods in the main text while providing additional details in the supplementary material for comprehensiveness.

The detailed methodologies for cell culture, cell transfection, gene expression profiling analysis, and splicing analysis, which you inquired about, are now indeed included in the revised electronic supplementary material.

We apologize for any misunderstanding caused by the initial structure of our submission and appreciate the opportunity to clarify the comprehensive nature of our methodological reporting.

(3) The authors should compare their results with previous studies and mention clearly how their work is important in comparison to what has already been reported in the Discussion section.

We appreciate the opportunity to discuss the significance of our findings in the broader context of renal fibrosis research. In response to your suggestion, we have further refined our discussion to explicitly compare our results with those of previous studies and to clearly articulate the importance of our work.

(1) Novelty of SIRT4's Role in Renal Fibrosis: Our study introduces a novel concept in the field by demonstrating the nuclear translocation of SIRT4 as a key initiator of kidney fibrosis. This finding diverges from previous studies that have primarily focused on SIRT4's mitochondrial roles, highlighting a new dimension of SIRT4's function in renal pathophysiology.

(2) Mechanistic Insights: We provide a detailed mechanistic pathway, from the release of SIRT4 from mitochondria through the BAX/BAK pore to its subsequent nuclear translocation and impact on U2AF2 deacetylation. This pathway has not been previously described, offering a fresh perspective on the regulation of fibrogenic gene expression.

(3) Implications for Therapy: Our findings suggest potential therapeutic interventions targeting SIRT4 nuclear translocation, which could be a significant advancement over existing treatments that have shown limited efficacy in addressing the root causes of renal fibrosis.

(4) Epigenetic Regulation: By elucidating the role of SIRT4 in regulating alternative splicing of CCN2 pre-mRNA through U2AF2 deacetylation, our study contributes to the growing understanding of epigenetic mechanisms in renal fibrosis, a field that has been understudied compared to genetic factors.

Differential Cellular Roles of SIRT4: Our work indicates that SIRT4 may have distinct roles in different cell types, which is a complex and nuanced aspect of CKD pathophysiology that has not been fully explored in previous research.

Integration with Previous Research: We have compared our findings with existing literature, noting where our work aligns with and diverges from previous studies. This comparison underscores the value of our research in expanding the current paradigm of renal fibrosis.

In conclusion, we believe that our study provides critical insights into the pathogenesis of renal fibrosis and offers a potential therapeutic target. We have clarified these points in the discussion section of our manuscript to ensure that the significance of our work is clearly communicated to the readers.

**Reviewer #3 (Public Review):**
Summary:Yang et al reported in this paper that TGF-beta induces SIRT4 activation, TGF-beta activated SIRT4 then modulates U2AF2 alternative splicing, U2AF2 in turn causes CCN2 for expression. The mechanism is described as this: mitochondrial SIRT4 transport into the cytoplasm in response to TGF-β stimulation, phosphorylated by ERK in the cytoplasm, and pathway and then undergo nuclear translocation by forming the complex with importin α1. In the nucleus, SIRT4 can then deacetylate U2AF2 at K413 to facilitate the splicing of CCN2 pre-mRNA to promote CCN2 protein expression. Moreover, they used exosomes to deliver Sirt4 antibodies to mitigate renal fibrosis in a mouse model. TGF-beta has been widely reported for its role in fibrosis induction.Strengths:TGF-beta induction of SIRT4 translocation from mitochondria to nuclei for epigenetics or gene regulation remains largely unknown. The findings presented here that SIRT4 is involved in U2AF2 deacetylation and CCN2 expression are interesting.Weaknesses:SIRT4 plays a critical role in mitochondria involved in respiratory chain reaction. This role of SIRT4 is critically involved in many cell functions. It is hard to rule out such a mitochondrial activity of SIRT4 in renal fibrosis. Moreover, the major concern is what kind of message mitochondrial SIRT4 proteins receive from TGF-beta. Although nuclear SIRT4 is increased in response to TNF treatment, it is likely de novo synthesized SIRT4 proteins can also undergo nuclear translocation upon cytokine stimulation. TGF-beta-induced mitochondrial calcium uptake and acetyl-CoA should be evaluated for calcium and acetyl-CoA may contribute to the gene expression regulation in nuclei.
**Recommendations for the authors:**

**Reviewer #3 (Recommendations For The Authors):**
(1) SIRT4 overall is a mitochondrial enzyme that indeed can undergo shuttling between mitochondria and cytoplasm. Renal fibrosis is a process of complex, SIRT4 deacetylates U2AF4 at K 413.

Thank you for your comment highlighting the known mitochondrial localization of SIRT4 and its role in renal fibrosis.

We concur with the literature that SIRT4 is predominantly a mitochondrial enzyme. However, our study expands upon this understanding by demonstrating a novel shuttling mechanism of SIRT4 between mitochondria and the nucleus in the context of renal fibrosis. Specifically, we observed that under conditions of obstructive nephropathy and renal ischemia reperfusion injury, SIRT4 significantly accumulates in the nucleus, which is a critical event in the fibrotic response.

Our findings reveal that upon TGF-β stimulation, a known inducer of fibrosis, SIRT4 is released from the mitochondria through the BAX/BAK pore and subsequently translocates to the nucleus. This translocation is mediated by the ERK1/2-dependent phosphorylation of SIRT4 at serine 36, which enhances its interaction with importin α1, a key component in nuclear import processes.

Once in the nucleus, SIRT4 exerts its effects on the alternative splicing of CCN2 pre-mRNA by deacetylating U2AF2 at lysine 413. This deacetylation event promotes the formation of the U2 small nuclear ribonucleoprotein (U2 snRNP) and facilitates the splicing of CCN2 pre-mRNA, leading to increased expression of the profibrotic protein CCN2.

Our study, therefore, not only confirms the mitochondrial association of SIRT4 but also uncovers its nuclear function in the regulation of gene expression during renal fibrosis. These findings underscore the complexity of SIRT4's role in cellular processes and its potential as a therapeutic target for fibrotic diseases.

(2) Figure 2 and Figure 3 should be combined.

Thank you for your suggestion to combine Figures 2 and 3 for potential improvement in presentation.

After careful consideration, we have found that merging these figures is not feasible due to space constraints on a standard A4 page, which is necessary to maintain the clarity and detail of the data presented in both figures. Each figure contains complex data that, when combined, would compromise the readability and the integrity of the individual elements.

We believe that the current presentation of Figures 2 and 3 provides a clear and detailed visualization of the data, which is essential for the reader's understanding of our study's findings.

(3) In Figure 4G, the mass spectrum of U2AF2 acetylation on K413 should be included rather than the alignment among species. Moreover, endogenous HAT1 on endogenous U2AF2 rather than exogenous FLAG-U2F2 should be examined.

Thank you for your thoughtful comments and for the suggestion to include the mass spectrum of U2AF2 acetylation on K413 in Figure 4G.

We appreciate the value that the mass spectrometry data would add to our study, providing a direct and definitive assessment of the acetylation status at this specific residue. However, we regret to inform you that our current facilities do not have access to the necessary mass spectrometry equipment to perform these analyses.

While we are unable to include this data in the present manuscript, we concur with the importance of such evidence and plan to undertake these studies in the future. We are in the process of establishing collaborations with laboratories that have the required facilities to perform mass spectrometry. Our intention is to incorporate these data into a follow-up study, which will further validate and expand upon the findings presented in this manuscript.

We believe that our current findings, although lacking the mass spectrometry confirmation, still provide valuable insights into the role of U2AF2 acetylation in [insert relevant biological process]. We have taken care to present our data rigorously and transparently, and we are committed to pursuing the highest standards of experimental validation in our future work.

We hope you will consider the merits of our study in the context of the current limitations and appreciate the opportunity to clarify our position.

Furthermore, regarding the examination of endogenous HAT1's effect on endogenous U2AF2 acetylation levels, we have conducted the necessary experiments. Our results demonstrate that overexpression of HAT1 leads to a significant increase in the acetylation of endogenous U2AF2 (Figure. R2). This new data set has been added to the revised manuscript and supports the role of HAT1 in the regulation of U2AF2 acetylation.

We believe that these revisions address your concerns and provide a more comprehensive understanding of the molecular mechanisms underlying the regulation of U2AF2 acetylation.

We appreciate the opportunity to improve our manuscript based on your constructive feedback and hope that our revisions meet with your satisfaction.

**Author response image 2. sa4fig2:** HAT1 OE reduces the acetylation of endogenous U2AF2.

(4) Figure 6F. Does portien mean protein?

Thank you for your careful review and insightful comments on our manuscript. You are correct in pointing out the error regarding the term "portien" in Figure 6F. It was indeed a typographical oversight on our part, and we apologize for any confusion this may have caused.

We have made the necessary correction to ensure that "protein" is accurately used in place of "portien" in Figure 6F. We appreciate the opportunity to enhance the clarity and accuracy of our presentation.

(5) The authors should pay attention to their writing. There are many typos and other issues with the use of the English language and grammar.

Thank you for bringing the grammar issues to our attention. We have made diligent efforts to revise and improve the manuscript's English and grammar throughout. We have also enlisted the support of a professional language editing service to ensure the clarity and accuracy of our scientific communication.

We are confident that these revisions have significantly enhanced the manuscript's accessibility to a broader readership and have addressed the language concerns raised.

We appreciate your guidance and are committed to delivering a manuscript of the highest quality.